# Geospatial Data Disaggregation through Self-Trained Encoder–Decoder Convolutional Models

**João Monteiro** [1,*], **Bruno Martins** [1], **Miguel Costa** [2] and **João M. Pires** [3]

1. IST/INESC-ID, Universidade de Lisboa, 1049-001 Lisboa, Portugal; bruno.g.martins@tecnico.ulisboa.pt
2. Vodafone, 1998-017 Lisboa, Portugal; miguel.costa2@vodafone.com
3. FCT/NOVA-LINCS, Universidade NOVA de Lisboa, 2829-526 Caparica, Portugal; jmp@fct.unl.pt
* Correspondence: joao.miguel.monteiro@tecnico.ulisboa.pt

**Abstract:** Datasets collecting demographic and socio-economic statistics are widely available. Still, the data are often only released for highly aggregated geospatial areas, which can mask important local hotspots. When conducting spatial analysis, one often needs to disaggregate the source data, transforming the statistics reported for a set of source zones into values for a set of target zones, with a different geometry and a higher spatial resolution. This article reports on a novel dasymetric disaggregation method that uses encoder–decoder convolutional neural networks, similar to those adopted in image segmentation tasks, to combine different types of ancillary data. Model training constitutes a particular challenge. This is due to the fact that disaggregation tasks are ill-posed and do not entail the direct use of supervision signals in the form of training instances mapping low-resolution to high-resolution counts. We propose to address this problem through self-training. Our method iteratively refines initial estimates produced by disaggregation heuristics and training models with the estimates from previous iterations together with relevant regularization strategies. We conducted experiments related to the disaggregation of different variables collected for Continental Portugal into a raster grid with a resolution of 200 m. Results show that the proposed approach outperforms common alternative methods, including approaches that use other types of regression models to infer the dasymetric weights.

**Keywords:** geospatial data disaggregation; dasymetric disaggregation; self-supervised learning; encoder–decoder neural networks; convolutional neural networks; deep learning

## 1. Introduction

Geospatial layers with statistical count data are widely available on a variety of subjects, such as economic activities or public health concerns. However, these data are often collected or released at an aggregated level, with insufficient spatial detail for some applications (e.g., aggregated data for districts or municipalities, hiding variations between smaller administrative units). In the aggregated form, the data are useful for broad-scale assessments. However, spatial variations tend to be over-smoothed, with local hotspots being masked. Alternatively, thin-grained information can better enable the formulation of informed hypotheses in the context of demographic, social, or environmental issues. It can also improve the analysis of the data through different partitions of space or in terms of their relation to particular terrain characteristics (e.g., analysing relations towards high-resolution data obtained through remote sensing or from volunteered geographic services).

High-resolution grids (i.e., geographically referenced lattices of cells, with each cell carrying a count value associated with its location) can be used to deliver geo-referenced statistical data. These representations have been extensively used for population distribution data [1–4]. Several projects have, for instance, reported on the construction of population grids through spatial disaggregation. The employed methods range in complexity, from simple mass-preserving areal weighting [5] to dasymetric weighting schemes that distribute the aggregated counts throughout the study region [6–8]. In most recent

approaches, the dasymetric surfaces are built using regression analysis [9–14] to combine different ancillary variables, for instance obtained from remote sensing. These variables can represent aspects such as land coverage, the location of buildings and road segments, or night-time light emissions. Still, most studies have used relatively simple regression algorithms (e.g., linear models [6,14] or, in some cases, ensembles of decision trees [4,12,13,15]), which process each grid cell independently of the others. Some authors have recently shown that more advanced learning approaches (e.g., models based on convolutional neural networks) can improve results on this task by taking advantage of important spatial context information [16–19]. However, these studies have used a small set of ancillary variables (i.e., mostly RGB satellite photos) to downscale data relative to the distribution of population, not addressing the application to other types of variables.

The training of the regression models that inform the disaggregation constitutes a particular challenge. This relates to the fact that the problem is ill-posed and we lack the ideal ground-truth data (i.e., data available at the target resolution in which the method is evaluated). To overcome having ground-truth data only at a coarse scale, most previous studies have taken the approach of training predictive models to estimate density, rather than directly estimating counts. One can, for instance, aggregate the ancillary variables (i.e., the regression covariates) according to the source administrative units, and train a regression model to predict density with a basis on data at that resolution [3,4]. After training, the model is used to create a density layer at the target resolution, followed by the dasymetric redistribution of the population counts. Aiming to avoid the aggregation of the ancillary variables to the source resolution, as well as the mismatch between the coarse resolution used for training and the target resolution, some recent studies have instead followed iterative self-training approaches [6,7]. In this case, disaggregated estimates are produced directly at the target resolution, iteratively refining results through the successive training of regression models over better estimates (e.g., starting from initial estimates produced through baseline disaggregation heuristics such as pycnophylactic interpolation). Some of these studies have tested different regression models for combining the ancillary data, although again mostly focusing on relatively simple approaches. Moreover, the authors have not explicitly studied the convergence properties of self-training approaches, e.g., by analysing the results at each iteration as a function of model hyper-parameters.

In this article, we report on a novel spatial disaggregation method that explores self-training together with deep neural networks for combining different types of ancillary data. We particularly use an encoder–decoder convolutional neural network, similar to those adopted in studies processing remote-sensing data for land coverage classification and/or image segmentation [20]. This specific regression approach can naturally capture contextual spatial information, allowing us to deal with the intrinsic characteristics of the spatial disaggregation task (e.g., complex relations between the ancillary variables and the associated spatial context). While previous studies have already used self-training methods in the context of spatial disaggregation, they have only used this approach together with simpler regression algorithms. On the other hand, previous studies that used similar deep learning methods, such as the regression approach for inferring the dasymetric weights, have important limitations (e.g., requiring source data already at a relatively high resolution), which are not present in the proposed approach.

Previous studies on self-training with deep neural networks have suggested that proper model regularization is an important concern. Some authors emphasized its importance in image classification tasks, stating that without adopting regularization within these frameworks, the models will learn to closely reproduce the original inputs, instead of generalizing towards the improved estimates that are required for self-training [21]. We therefore followed this idea on the task of spatial disaggregation. We experimented with different regularization strategies that promote heterogeneity in the results, or promote equivariances to spatial transformations, such as rotations in the inputs. Moreover, in an attempt to better understand the self-training process in this particular application, we analysed the convergence of the estimates that are iteratively produced as a func-

tion of two aspects, namely (a) when optimizing the model with different loss functions and regularization strategies, and (b) when replacing the neural regression model with alternative algorithms.

The proposed approach was evaluated on the disaggregation of data originally available at the level of coarse administrative regions, such as NUTS III units, into high-resolution grids with a resolution of 200 × 200 m per cell. We specifically considered socio-economic variables relative to the territory of Continental Portugal, comparing the proposed approach against several common alternative methods. These include simple non-regression baselines (e.g., mass-preserving areal weighting, pycnophylactic interpolation, or dasymetric weighting, leveraging population density as the weights), as well as approaches based on other regression algorithms (e.g., linear regression or random forests), with or without the combination of self-training. Existing studies in the literature have considered different disaggregation tasks and different sets of ancillary variables, making direct comparisons with published results difficult. As such, to comparatively assess the proposed approach, we emulated previous techniques through different parameterizations of the proposed approach (e.g., using different regression algorithms, or looking at the results of a single iteration of self-training). This way, we performed consistent evaluation experiments with the same tasks and datasets. The obtained results show that self-training indeed leads to improved performance with different types of regression algorithms. Learning the dasymetric weights through the encoder–decoder network also contributed to slight improvements, and the proposed regularization strategies contributed to better convergence properties.

In brief, the main contributions of this article can be summarized as follows:

- We proposed a novel spatial disaggregation method, based on self-training an encoder–decoder convolutional network as the regression model that combines the sources of ancillary information.
- We compared different loss functions and model regularization strategies for training the neural network in an attempt to increase robustness to outliers, promote heterogeneity in the results, and promote the equivariance to simple spatial transformations in the input data, such as rotations.
- We evaluated the proposed approach on the disaggregation of statistical data relative to the territory of Continental Portugal. We also compared the disaggregation errors against those obtained with our implementation of popular alternative methods. The experimental results show that our approach outperforms common alternative methods, in some cases by a significant margin.
- We analysed the convergence properties of the self-training framework under different settings, concluding on its contribution for improved result quality.

The rest of this article is organized as follows: Section 2 presents related work in the area of geospatial data disaggregation. Section 3 describes the proposed disaggregation method, while Section 4 details the experimental setup that was used. Section 5 presents the results obtained from the application of the proposed approach to the disaggregation of data for Continental Portugal. Finally, Section 6 presents our main conclusions and highlights possible directions for future work.

## 2. Related Work

This section begins by explaining classical spatial disaggregation methods, and it then discusses more recent advances and practical applications.

### 2.1. Seminal Methods for Spatial Data Disaggregation

The simplest spatial disaggregation method is mass-preserving areal weighting, which assumes a homogeneous distribution of the data throughout each source zone [5]. This technique redistributes the aggregated data based on the proportion of each source zone (e.g., the administrative units at which the data is originally available) that overlaps with each target zone (e.g., the cells in the target high-resolution raster representation). While

this procedure ensures that the total counts from the source data remain unchanged, it is based on the often incorrect assumption that the phenomena of interest are evenly distributed across the source zones.

Pycnophylactic interpolation is a refinement over mass-preserving areal weighting, assuming a degree of spatial auto-correlation in the data distribution [22]. This method starts by applying the mass-preserving areal weighting procedure, afterward smoothing the estimated values for the target grid cells by replacing them with the average of their neighbors. The predicted values for all cells within each source zone are then compared with the actual values (i.e., the aggregated counts in the source data), and adjusted to meet the condition of mass-preservation. The algorithm continues until there are no significant changes over the cell values from the previous iteration.

Although pycnophylactic interpolation no longer assumes a uniform distribution for the data, it makes no attempt at modeling this distribution as a function of particular properties associated with the target zones. Dasymetric weighting schemes are instead based on first creating weighted surfaces, capturing the relevant properties of the target zones, to distribute the source data accordingly [8]. The weights are usually determined from the analysis of one or multiple spatial layers (e.g., water bodies, building footprints, etc.) according to rules that relate the ancillary variables to the expected counts. While some schemes use simple binary masks built from land-coverage data, other approaches rely on expert knowledge and manually-defined rules. For instance, in the tests that are reported later in this paper, we explored the use of a weighted surface with values that are proportional to the population density, for disaggregating other variables. More recent methods leverage regression analysis and machine learning to automatically learn the dasymetric weights [4,6,23,24].

### 2.2. Dasymetric Weighting Based on Regression Analysis

In the context of the WorldPop project, Stevens et al. [4] developed a technique for creating gridded predictions of population density, with a resolution of approximately $100 \times 100$ m. The proposed approach uses a random forest algorithm in order to predict population density from aggregated remotely-sensed and geospatial data. In brief, the method corresponds to a multi-stage estimation technique, which first tunes the number of covariates that are randomly selected for splitting at each tree node. In a second phase, a covariate selection process is conducted in order to reduce the number of total features and accelerate per-pixel predictions. More specifically, covariate importance is inferred from a fitted model and used to remove from the list of potential features the ones with a score of zero. The procedure is iterated until only positive importance scores remain for every covariate included in the modeling process. Each tree of the resulting random forest is used to produce estimates for each pixel, and the resulting map is obtained from averaging the different results. For evaluation purposes, Stevens et al. used three case studies for the countries of Cambodia, Vietnam, and Kenya. First, the density maps were computed from aggregated data available at coarse regions (i.e., divisions or provinces). Then, they were used in a standard dasymetric mapping approach for obtaining the population counts at each pixel. Finally, the pixel values within each of the finer census units (i.e., village or sub-locations) were summed and compared with the corresponding known counts through metrics such as the Mean Absolute Error (MAE) or the Root Mean Squared Error (RMSE). The authors concluded that their method outperformed several competitors, such as the products produced within the GRUMP [25], GPW [2], or the AfriPop/AsiaPop [15,26] projects, in the reported metrics.

In another recent study exploring the use of regression analysis to infer dasymetric weights, Cheng et al. reported on the disaggregation of census data for China into a raster grid with a resolution of $1 \times 1$ km per cell for each month in 2015 [27]. The authors combined environmental information and mobile phone positioning data as the ancillary variables that were used to infer the dasymetric weights. They also proposed a hybrid inference approach, combining random forests with area-to-point kriging. The random

forest model is trained with data at the town level, aggregating the ancillary data (i.e., taking the mean values per town as the independent variables) and using the population density as the target variable. The model is then used to produce population estimates for the target cells, which are re-aggregated to the town level for computing the areal residuals for each town. The area-to-point kriging model finally uses this information to adjust the random forest predictions under the assumption that the sum of the encompassing residuals at the pixel level should match the town's residual. The pixel residuals are calculated through a weighted linear combination of the residuals of neighboring towns. The authors compared the disaggregated values obtained with their method against the ones obtained using gridded population products such as WorldPop or GPW, and with the application of the random forest model without subsequently adjusting the estimates with area-to-point kriging. The proposed approach achieved the best results in the terms of the $R^2$ between predicted and real data. Moreover, it efficiently addressed a problem that the authors observed with the random forest model, which lead to the over-estimation of population values.

### 2.3. Spatial Data Downscaling Using Self-Training

Although dasymetric weighting methods can effectively use ancillary data, the training of regression models to infer the dasymetric weights constitutes a particular challenge. To overcome the mismatch between the target zones and the coarse resolution at which the source data is available (i.e., the regression models are usually trained for estimating density from ancillary data aggregated to the source zones), some studies have proposed the use of self-training approaches that operate directly at the target resolution, progressively refining estimates for the target values.

Malone et al. [9] presented a method named dissever for downscaling soil organic carbon data through the use of ancillary variables available as fine resolution grids. The dissever algorithm starts with an initialization phase, where a re-sampling procedure is used to transfer the data from the source to the target regions. This operation is followed by the application of a regression model (i.e., a generalized additive model) to predict initial estimates from the set of covariates. Then, updates are iteratively made to the estimates. First, they are aggregated to the level of source zones, compared with the available values, and adjusted accordingly. Then, a new generalized additive model is applied to predict updated values for all the grid cells.

Subsequent adaptations of the dissever procedure have been made by Monteiro et al. [6,7], which considered spatial disaggregation instead of the downscaling of non-additive variables. In the first study, the authors presented a general disaggregation methodology (which is also utilized in the present article, with minor adaptations) that combines pycnophylactic interpolation with dasymetric mapping, and used it for disaggregating Portuguese socio-economic variables. The authors experimented with methods such as linear or generalized additive regression to combine the different sources of ancillary information. In the second study, the same methodology was applied for disaggregating historical census data for the territories of the Netherlands, Belgium, and Great Britain. However, in this second study, the authors leveraged more advanced machine learning regression algorithms, such as ensembles of decision trees and a neural network based on the LeNet-5 architecture [28]. The good results reported in both studies from Monteiro et al. motivated the experimentation with more advanced neural network architectures, such as the one used in the present article.

### 2.4. Deep Learning for Population Mapping

Despite the potential of deep learning methods for geospatial data analysis applications, only a few previous studies have explored their use for tasks related to spatial disaggregation. Three of these exceptions are the studies from Tiecke et al. [29], Robinson et al. [16], and Jacobs et al. [18].

The study from Tiecke et al. describes a spatial disaggregation method for producing population maps. It first leverages a convolutional neural network based on the SegNet [30] and FeedbackNet [31] architectures for segmenting individual buildings in high-resolution satellite imagery. The building footprints are then used as a mask for proportionally allocating population counts. Specifically, the census counts are distributed proportionally to the fraction of built-up area within cells of $30 \times 30$ m.

Robinson et al. outlined a deep learning approach for producing high-resolution forecasts of spatial variables such as population counts, leveraging satellite imagery as ancillary information. The authors used the images to produce a high-resolution population grid for the territory of the United States at a resolution of $0.01° \times 0.01°$ (approximately 1 km$^2$). They considered five different CNN architectures based on the well-known VGG model, which were trained on data concerning the year of 2000. The networks specifically receive satellite imagery for each of the target areas, and output the corresponding population forecasts. Validation was performed using two approaches, namely a quantitative approach that compares the estimates aggregated at the county level to US Census population projections for the year of 2010, and a qualitative approach based on interpreting the predictions in terms of the input images.

As an alternative to traditional dasymetric approaches, Jacobs et al. aimed to estimate density functions without labels directly associated with the computed results. The authors introduced a novel neural network component, called the regional aggregation layer (RAL), which aggregates the pixel-level estimates to a coarser extent during the training stage. At these new regions, aggregated values are available as supervision signals, allowing one to train an end-to-end CNN model based on the U-Net architecture [20]. The evaluation was conducted through tests related to the estimation of population density, leveraging the US Census data and high-resolution satellite imagery. The model using the RAL component achieved better MAE results when compared to a baseline approach which instead computes pixel-level errors under the assumption of a uniform distribution across the corresponding regions. Further testing their approach, the authors emphasized that it could also be used to redistribute population counts using the computed densities for dasymetric mapping. They specifically experimented with the incorporation of population and housing density predictions to this end, although they did not quantitatively evaluate the results due to the lack of high-resolution counts. Despite the good results, Jacobs et al. pointed out a limitation related to the fact that their approach requires the different regions, for which the reference data is available, to be fully contained in an input patch to the neural network. This means that the proposed procedure can only be used to downscale data that are already available at a relatively high resolution.

*2.5. Overview and Discussion on the Related Work*

Table 1 presents an overview of the previous studies that were surveyed in this section. Seminal approaches used simple techniques that assume a uniform distribution of the target variable across the source areas [5], or derivations that take into account spatial auto-correlation [22]. Dasymetric mapping methods instead disaggregate the source counts according to weights derived from ancillary data [8]. Most recent disaggregation methods use some form of regression analysis for finding the best set of weights for redistributing the aggregated data [4,6,7,27].

**Table 1.** The spatial disaggregation approaches that have been surveyed.

| Study | Self-Training | Deep Learning | Task | Method |
|---|---|---|---|---|
| Goodchild et al. [5] | No | No | Disaggregate population counts | Mass-preserving areal weighting |
| Tobler and Waldo [22] | No | No | Disaggregate population counts | Pycnophylactic interpolation |
| Qiu et al. [8] | No | No | Disaggregate population counts | Dasymetric mapping |
| Stevens et al. [4] | No | No | Estimate population density | Random forests |
| Cheng et al. [27] | No | No | Disaggregate population counts | Random forests and area-to-point kriging |
| Malone et al. [9] | Yes | No | Downscale soil organic carbon data | Generalized additive regression |
| Monteiro et al. [6] | Yes | No | Disaggregate socio-economic data | Linear and generalized additive regression |
| Monteiro et al. [7] | Yes | Yes | Disaggregate population counts | Ensembles of decision trees and LeNet-5 |
| Tiecke et al. [29] | No | Yes | Disaggregate population with building footprints | SegNet and FeedbackNet |
| Robinson et al. [16] | No | Yes | Forecasting population counts | Architectures derived from VGG-Net |
| Jacobs et al. [18] | No | Yes | Estimate population density | Architecture derived from U-Net |
| Our approach | Yes | Yes | Disaggregate socio-economic data | Architecture derived from U-Net |

The approach described in this article combines and extends some ideas from previous studies. It uses an encoder–decoder neural model, such as Jacobs et al. [18], together with a self-training procedure similar to that of Monteiro et al. [6,7], which can use one of the seminal disaggregation procedures for initialization.

## 3. The Proposed Geospatial Data Disaggregation Approach

The disaggregation method proposed in this paper combines pycnophylactic interpolation and regression-based dasymetric mapping, following the general ideas that were originally advanced by Malone et al. [9], and subsequently adapted by Monteiro et al. [6,7]. As illustrated in Figure 1, a baseline disaggregation heuristic (e.g., pycnophylactic interpolation) can act as an initial teacher model in our approach, creating initial estimates from the aggregated data. The results are then iterated through a self-training procedure, in which a student regression model infers the distribution of the target indicator, as given by the teacher, from a set of ancillary variables. The student model is then used to produce new estimates, which act as the teacher supervision in a next iteration. In more detail, the different steps in the proposed procedure are as follows:

1. Produce a vector polygon layer for the aggregated information, by associating the source counts with the corresponding regions;
2. Create a raster representation for the study area, containing disaggregated estimates obtained from the layer in Step 1 through a baseline heuristic, such as pycnophylactic interpolation [22];
3. Train a regression model to infer the results from Step 2 from the ancillary information available as gridded rasters. Each ancillary dataset is normalized to the target resolution, through averaging/summing cells in the cases where the original raster had a higher resolution, or through bicubic interpolation in the cases where the original raster had a lower resolution. After training, the regression model is used to produce new disaggregated values;
4. Proportionally adjust the values returned by the regression model from the previous step for all cells within each source zone, so that each source zone's total in the target raster is the same as the total in the original vector polygon layer;
5. Steps 3 and 4 are repeated, aiming to adjust the disaggregated estimates until the estimated values converge (e.g., based on a stopping criterion), or until reaching a maximum number of iterations.

Given that the disaggregated estimates are produced in a self-supervised manner, through which the result of each iteration is used to inform the training of the following, having an appropriate stopping criterion is fundamental. Since we do not have a ground-truth dataset, it is not possible to retain a specific iteration based on a typical evaluation of performance improvement. Instead, in the experiments reported in this paper, we computed a fixed number of iterations (in our case, 30), and also tested the standard deviation of the values within the computed map as a heuristic for inferring the best iteration. Results showed that the disaggregated output should have high heterogeneity, and that the best iteration within our experimental setup is usually the one whose disaggregated map has the highest standard deviation in the target values.

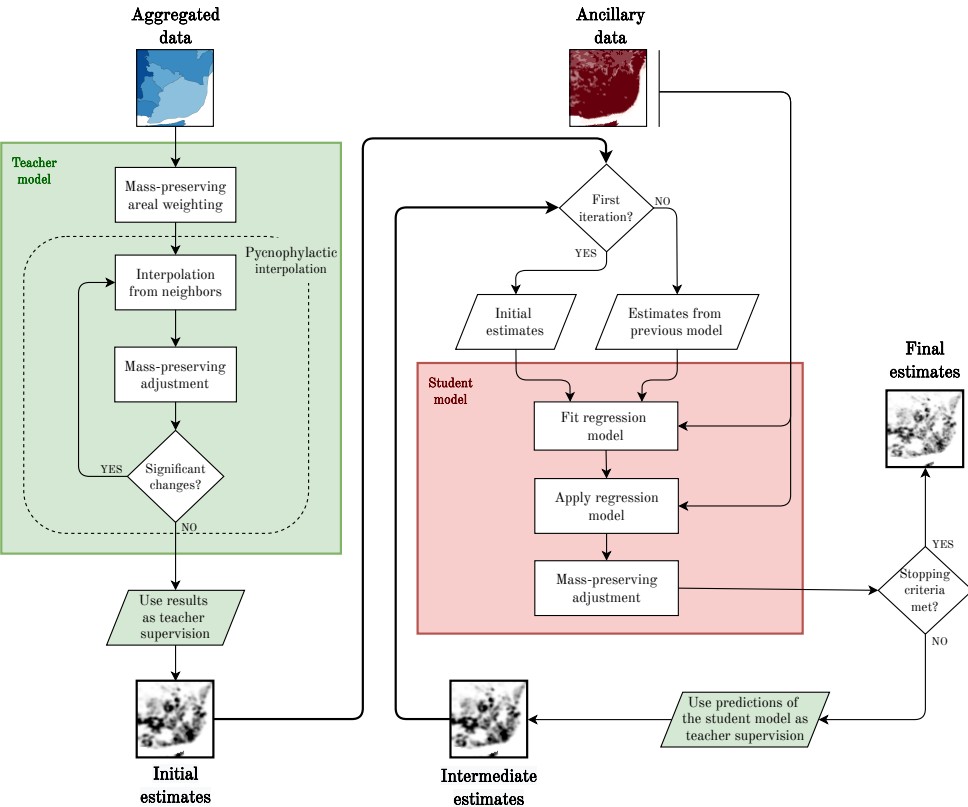

**Figure 1.** The different steps involved in the proposed disaggregation method, considering the use of initial disaggregation weights based on pycnophylactic interpolation.

The following subsections detail the aforementioned general approach. In particular, Section 3.1 presents the heuristics that we tested for producing the initial estimates, while Section 3.2 describes in detail the encoder–decoder CNN architecture that we tested as the regression algorithm for combining the different sources of ancillary data. Section 3.3 details our self-training framework, presenting its similarities to teacher/student approaches from the literature [21]. Finally, Section 3.4 describes the use of (i) a training objective tailored for dealing with the intrinsic characteristics of the data, (ii) a loss penalty for homogeneous patches, and (iii) a training strategy that promotes equivariance to spatial transformations on the input data.

### 3.1. Initial Disaggregation Estimates

We specifically experimented with the application of different heuristics for producing the initial estimates. Table 2 details the tested methods, ranging from pycnophylactic interpolation to a weighted interpolation (WI) strategy which disaggregates the data proportionally to the population density.

The estimates obtained through pycnophylactic interpolation take into account the property of spatial auto-correlation, which states that regions close to each other tend to have similar values. However, the method does not enforce other properties about the distribution of the target variable, and often leads to over-smooth results. The alternative weighted interpolation method produces estimates that reflect different densities across the territory, but the variable to be disaggregated may not exactly follow the population distribution. Taking this into account, we complemented the WI procedure with a smoothing process, this way promoting auto-correlation over the results. In particular, the values are smoothed through one iteration of the pycnophylactic interpolation procedure, i.e., by replacing all the cells with the average of their neighbors.

**Table 2.** The different heuristics tested for producing the initial estimates.

| Model | Description |
|---|---|
| Mass-preserving areal weighting | The original method from Goodchild et al. [5] |
| Pycnophylactic interpolation | The original method from Tobler and Waldo [22] |
| Weighted interpolation (WI) | Disaggregation proportionally to population density |
| Smooth weighted interpolation | Smoothing WI estimates by averaging neighbors |

*3.2. The Encoder–Decoder Convolutional Architecture*

Standard regression algorithms (e.g., linear regression or random forests) process each grid cell independently of the others to predict the disaggregated values. The values for each cell in the raster grid are used as the features of the model. These methods are thus limited, since they do not consider that each cell may be directly influenced by its neighbors. To overcome this problem, we experimented with a Convolutional Neural Network (CNN) architecture that processes input patches of fixed dimensions. In this case, the input layer of the CNN is a three-dimensional array of size $h \times w \times d$, where $h$ and $w$ are spatial dimensions of height and width, and $d$ is the feature dimension corresponding to the number of different ancillary variables.

We specifically used a CNN model based on an encoder–decoder architecture. Although the proposed method is generalizable to any architecture, we experimented with an adapted U-Net model, given that these architectures typically achieve good results on problems related to image segmentation [20]. The adapted architecture is described next, and illustrated in Figure 2.

- A contracting path consists of a sequence of four blocks, each combining convolution and pooling operations. Specifically, these blocks have two standard convolution operations followed by ReLU activation functions, using $3 \times 3$ convolution kernels, a stride of one, and zero padding. They also feature a final $2 \times 2$ max-pooling operation after the second convolution. All blocks in the contracting path adopt the same structure, each of them halving the spatial dimensionality of the input while at the same time doubling the number of feature channels.
- A middle block consists of two convolutional operations, similar to those in the contracting path and without considering the final max-pooling operation. The representation resulting from the middle block is also processed through a dropout operation, which drops out random units during training to reduce over-fitting and improve generalization.
- An expansive path consists of a sequence of four blocks, each of them matching one of the blocks in the contracting path. These blocks perform upsampling of the feature maps through a $2 \times 2$ up-convolution operation, concatenating the results with the feature maps from the corresponding blocks in the contracting path. The results from the concatenation are then further processed by two $3 \times 3$ convolution operations, similar to those used in the blocks from the contracting path.
- Different from the original U-Net architecture, we combine the feature maps resulting from the final block in the expansive path with the values directly obtained from the input patches through a shortcut (skip) connection. By doing so, we take direct advantage of the input ancillary variables, which contain rich information associated with each cell.
- The entire model is trained end-to-end, connecting the final layer with a loss function customized to our particular problem.

It is important to notice that large study regions cannot be analysed directly as a single input to the U-Net model due to the limited memory of the graphics processing units used for model training. A common solution involves extracting smaller sub-patches by sliding a window over the input, afterward processing these patches individually in order to generate new patches with the predictions. The resulting patches are finally stitched together in order to form a complete raster with the predictions. In the case of overlaps, the

resulting predictions can be obtained from averaging the results at the individual patches. Figure 3 illustrates this general procedure, which we also adopted.

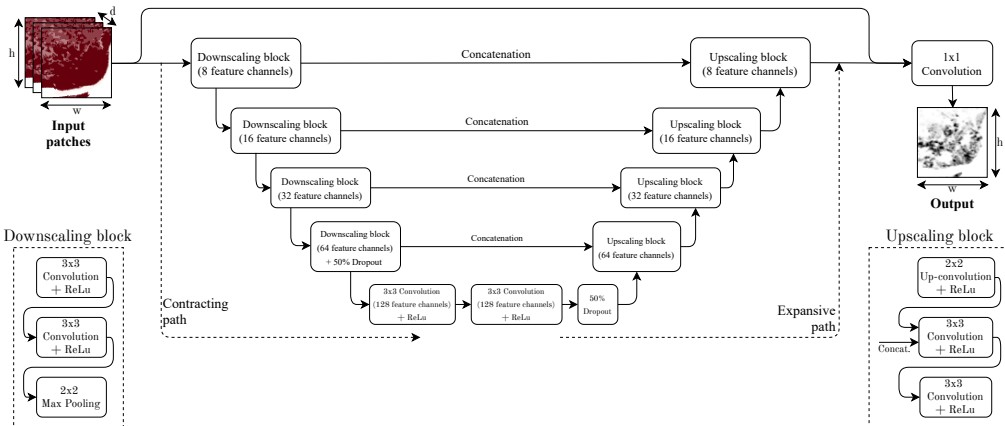

**Figure 2.** The adapted encoder–decoder deep neural network architecture.

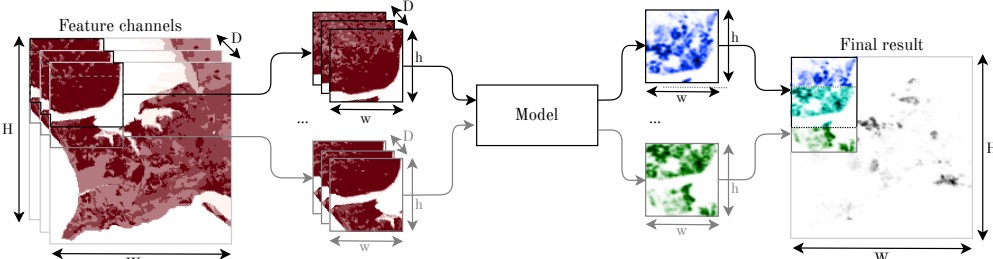

**Figure 3.** Illustration of the strategy to iteratively convert large spatial inputs into multiple patches.

### 3.3. Self-Training for Spatial Data Disaggregation

Spatial disaggregation methods that involve machine learning are particularly challenging in terms of the training process. For instance, in our tests, we disaggregate counts to a $200 \times 200$ m raster grid through a combination of ancillary data that is available at that resolution. However, at the level of the target cells, there are often no ground-truth data (e.g., high-resolution data for a given region used to train a model that would then be applied to a different region) concerning the variable to be disaggregated, with counts instead as being available in the aggregated form. Some studies have addressed this problem by comparing, during training, the produced estimates re-aggregated to zones at which reference data are available, such as neighborhoods or civil parishes [18]. However, this masks the variations at the level of raster cells. Instead, we base our training process on a self-training framework [6,7,9] which can be used in combination with different types of regression algorithms (e.g., encoder–decoder CNNs).

Our disaggregation method has strong similarities with self-training approaches that have been recently outlined in the literature, such as the teacher/student framework used for image classification by Xie et al. [21]. We use the initial estimates produced with a relatively simple disaggregation heuristic (e.g., pycnophylactic interpolation) that can be seen as a teacher model. These results are then used to train a regression model (i.e., the student), whose predictions are iteratively refined until some convergence criteria are met. Like in the study from Xie et al., in which the authors performed model regularization through the injection of noise, dropout, stochastic depth, and data augmentation, we also applied regularization strategies. Specifically, we used penalties based on the standard deviation of the produced results, as well as on results computed from transformed versions of the input data. The CNN used in our tests also considers built-in regularization in the form of dropout in some of its layers.

### 3.4. Model Training

The selection of an appropriate loss function is critical for effective model training, quantifying the error that is back-propagated and used to adjust the model parameters. The most typical regression loss functions are based on the Mean Absolute Error (MAE) or the Root Mean Square Error (RMSE), respectively, presented in Equations (1) and (2). While the MAE penalizes the errors linearly, the RMSE penalizes them quadratically, thus being more sensitive to large differences.

$$\text{MAE}(y, \hat{y}) = \frac{\sum_{i=1}^{n} |y_i - \hat{y}_i|}{n}. \tag{1}$$

$$\text{RMSE}(y, \hat{y}) = \sqrt{\frac{\sum_{i=1}^{n} (y_i - \hat{y}_i)^2}{n}}. \tag{2}$$

In Equations (1) and (2), $y_i$ corresponds to a ground-truth value, $\hat{y}_i$ corresponds to a predicted value, and $n$ is the number of instances. Despite being commonly used, both losses have some known problems. For example, the MAE loss produces a constant gradient, even for small values. On the other hand, the RMSE loss can over-penalize extreme values in the data. One possible alternative is the Huber loss, which tries to overcome the aforementioned issues through an interpolation from quadratic to linear penalization. We specifically tested the standard Huber loss, as defined in Equation (3).

$$\text{HuberLoss}_\delta(y, \hat{y}) = \begin{cases} \frac{1}{2}(y - \hat{y})^2, & \text{for } |y - \hat{y}| \le \delta \\ \delta(|y - \hat{y}| - \frac{1}{2}\delta), & \text{otherwise.} \end{cases} \tag{3}$$

In Equation (3), $\delta$ is the threshold that separates the different behaviors of the loss function. In other words, the Huber loss has a quadratic behavior when the error is below $\delta$, and a linear behaviour otherwise. This way, large differences will have a smaller impact on the final loss value.

Further trying to improve disaggregation results, we complemented the training loss with components that penalize predictions that are less plausible in our scenario, at the same time benefiting the ones that contain expected characteristics in the produced estimates. In this context, as discussed in Section 1, we know that most variables are rarely uniform when disaggregated to high-resolution grids. Taking this into account, and also knowing that the standard deviation of the estimated disaggregated results can be used as a proxy for their variation, we incorporated this metric as a component within our overall training loss.

In addition, we also took advantage of the fact that the absolute orientation of the ancillary data used in our disaggregation approach is irrelevant. In other words, the use that is made of the elements in the input images should not change if they suffer geometric transformations such as rotations. Within CNNs, equivariance to transformations of the inputs can be approximated by using data augmentation. If a model has enough capacity and has seen training examples after the application of a sufficient number of transformations, it will learn to be invariant to these factors. However, instead of significantly increasing the training data, it can be more interesting to directly promote these equivariances during model training by changing the loss function. We therefore incorporated a loss term to maximize the agreement between original and transformed image representations. This solution forces the CNN to learn similar discriminative representations to both input versions, thus improving generalization. The general procedure is illustrated in Figure 4, and detailed next:

1. First, we apply a random transformation to the input patch, such as flipping over an axis or performing a rotation;
2. Then, we apply a forward pass with the model over both the original and the transformed patch;

3.    Finally, we apply the inverse transformation to the output computed when using the transformed patch and produce the final results by averaging both outputs.

The model training procedure is then adapted for measuring not only the difference between the produced estimates against the target values, but also the difference between the two predicted outputs (i.e., obtained by processing the original patch and the corresponding transformed version).

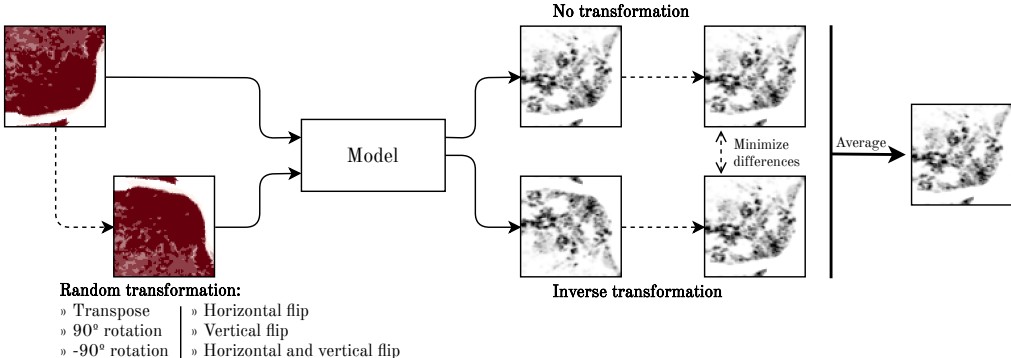

**Figure 4.** Illustration of the strategy used to encode equivariances to geometric transformations.

The corresponding global loss is presented in Equation (4), and it is comprised of three parts: (i) $\mathcal{L}\text{Cellwise}(y, \text{avg}(\hat{y}_1, \hat{y}_2))$ corresponds to a loss (e.g., the Huber loss) between the target and predicted values at the level of individual cells within the patches, (ii) $\mathcal{L}\text{Compatibility}(\hat{y}_1, \hat{y}_2)$ corresponds to a loss (e.g., the Huber loss) between the two versions of predicted patches, as detailed in Figure 4, and (iii) a $\mathcal{L}\text{Homogeneity}(\hat{y}_1, \hat{y}_2)$ corresponds to the regularization term defined in Equation (5), derived from the standard deviation of the values computed from averaging the two predicted patches (i.e., $\text{std}(\text{avg}(\hat{y}_1, \hat{y}_2))$). The three components are weighted by the parameters $w_1$, $w_2$, and $w_3$, which control their relative importance.

$$\begin{aligned}
\mathcal{L}\text{Global}(y, \hat{y}_1, \hat{y}_2) = & w_1 \mathcal{L}\text{Cellwise}(y, \text{avg}(\hat{y}_1, \hat{y}_2)) + \\
& w_2 \mathcal{L}\text{Compatibility}(\hat{y}_1, \hat{y}_2) + \\
& w_3 \mathcal{L}\text{Homogeneity}(\hat{y}_1, \hat{y}_2).
\end{aligned} \tag{4}$$

$$\mathcal{L}\text{Homogeneity}(\hat{y}_1, \hat{y}_2) = \frac{1}{1 + \text{std}(\text{avg}(\hat{y}_1, \hat{y}_2))}. \tag{5}$$

## 4. Experimental Setup

We evaluated the proposed spatial disaggregation approach with tests involving the disaggregation of socio-economic data pertaining to Continental Portugal and its administrative units. We considered the four variables detailed in Table 3, using data for the year of 2019. These consist of information concerning (i) the overall amount of withdrawals from automated teller machines, in thousands of euros, between January and December (*withdrawals—all year*), (ii) a subset of the previous for the summer period, i.e., for the months of June to September (*withdrawals—summer*), (iii) a subset of the first for the winter period, i.e., for the months of December to March (*withdrawals—winter*), and (iv) the number of live births by place of residence of the mother (*live births*). It is important to notice that the reason for using different sub-datasets concerning the amount of withdrawals relates to the aim of assessing two aspects. First, to which extent our disaggregation method could detect patterns depending on the months that are considered, for example resulting from the movement of the population. Second, to which extent the model would be able to explore all the ancillary variables it receives, and more specifically the ones that differ in both temporal periods, namely the information on night-time lights.

We used different versions of this specific ancillary dataset, corresponding to averaged values for the different months that are considered in the variables to be disaggregated.

**Table 3.** The datasets used in our experimental evaluation.

| Dataset | Source | Year | Resolution | Type |
|---|---|---|---|---|
| Withdrawals—All year | National Institute of Statistics (INE) | 2019 | Municipalities | Aggregated |
| Withdrawals—Summer | National Institute of Statistics (INE) | 2019 | Municipalities | Aggregated |
| Withdrawals—Winter | National Institute of Statistics (INE) | 2019 | Municipalities | Aggregated |
| Live births | National Institute of Statistics (INE) | 2019 | Civil Parishes | Aggregated |
| Terrain development | Global Human Settlement project | 2015 | $38 \times 38$ m | Ancillary |
| Population density | Global Human Settlement project | 2015 | $250 \times 250$ m | Ancillary |
| Nighttime lights | VIIRS Nighttime Lights dataset | 2016 | $450 \times 450$ m | Ancillary |
| Land coverage | Corine Land Cover dataset | 2018 | $100 \times 100$ m | Ancillary |
| Human settlements | Copernicus Land Monitoring Service | 2015 | $10 \times 10$ m | Ancillary |

The evaluation process is particularly challenging since there are no ground-truth data with which we can compare results at the level of thin-gridded cells. Most information on socio-economic variables is only available at a coarse resolution, limiting us to infer the quality of the results only for aggregated areas. Our evaluation procedure is illustrated in Figure 5. We first disaggregate the information available at the level of large territorial divisions (i.e., the 24 NUTS III regions concerning the Continental Portugal territory), producing high-resolution estimates at the raster level. Then, we evaluate the results by re-aggregating them to the level of municipalities (i.e., taking the sum of the values from all raster cells associated with each municipality). At these intermediary regions, the results are compared against the known values that are available for the 278 municipalities. Notice that, in a real application, we would start the disaggregation from the higher resolution (e.g., municipalities or civil parishes), which is not possible for our experiments because we would not have the intermediate ground-truth for evaluation.

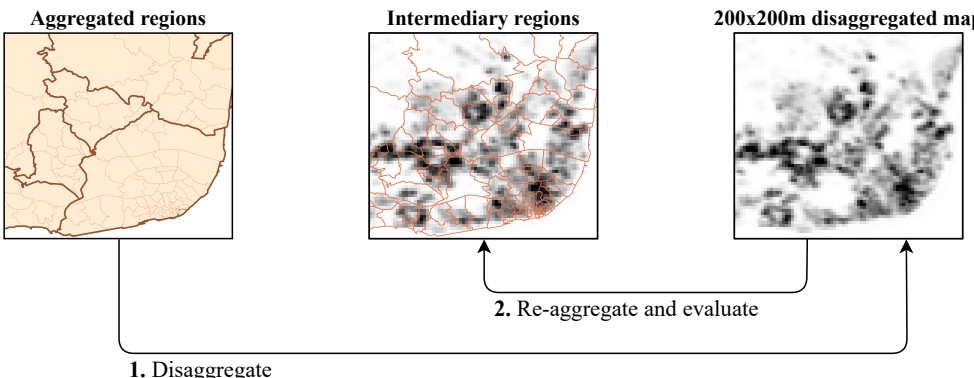

**Figure 5.** Illustration of the strategy used to evaluate the disaggregated estimates.

The following subsections describe the sources of ancillary information that have been explored (Section 4.1), which are also summarized in Table 3, together with implementation details (Section 4.2). Then, we present the evaluation metrics that were used to evaluate the different methods (Section 4.3).

### 4.1. Ancillary Datasets

The ancillary datasets that we used in our experiments are detailed in Table 3. In particular, the information regarding terrain development was obtained from the Global Human Settlement (GHS (http://ghsl.jrc.ec.europa.eu/datasets.php, accessed on 14 September 2021)) project [32–35], which aims to map the distribution and density of the world's built-up areas. This project analyzed Landsat imagery related to the periods of 1975, 1990, 2000, and 2013–2014 to quantify built-up structures in terms of their location and density.

The project specifically makes available raster grids with a resolution of 38 m per cell which express the distribution of built-up areas as the proportion of occupied footprint in each cell. In this article, we used the GHS built-up presence grid related to the year of 2015.

Besides the terrain development raster, the GHS project also provides population density grids for the same years, with a resolution of 250 m per cell. The methodology for building these grids is based on raster-based dasymetric mapping, using the GHS built-up presence to restrict and refine the population information available through the Gridded Population of the World (GPW (http://beta.sedac.ciesin.columbia.edu/data/collection/gpw-v4, accessed on 14 September 2021)) dataset. This other population grid was constructed from national or subnational input units (i.e., from low-level administrative units from the different countries), also through a spatial disaggregation approach. We specifically used the GHS population density layer referring to the year of 2015.

As for the ancillary information regarding night-time light emissions, the publicly available VIIRS Nighttime Lights—2016 dataset (http://ngdc.noaa.gov/eog/viirs/download_dnb_composites.html, accessed on 14 September 2021) was used, which is maintained by the Earth Observation Group of the NOAA National Geophysical Data Center. We specifically used the global cloud-free composite of VIIRS night-time lights, which was generated with VIIRS day/night band (DNB) observations collected on nights with zero moonlight. Cloud screening was performed on this dataset by detecting clouds in the VIIRS M15 thermal band. However, the product has not been filtered to subtract background noise or to remove light detections associated with fires, gas flares, volcanoes, or aurora. The raster data which we used, available at a resolution of $450 \times 450$ m per cell, consist of floating-point values calculated by averaging the pixels deemed to be cloud-free.

Since land coverage information is expected to correlate with the distribution of socio-economic variables [36], we also leveraged the standard Corine Land Cover (CLC) product (http://land.copernicus.eu/pan-european/corine-land-cover, accessed on 14 September 2021) for the year of 2018, available at a resolution of 100 m. This dataset is based on satellite images as the primary information source, and the technical details are presented in the report by Heymann and Bossard [37]. The 44 different classes of the 3-level Corine nomenclature, considered in the original product (e.g., classes for water bodies, artificial surfaces, agricultural areas, etc.), were converted into a real value in the range $[0, 1]$, which encodes how developed the territory corresponding to a given cell in a simple dasymetric distribution is that corresponds to a class-percent method. More specifically, cells with the class water bodies were assigned the value of zero, cells corresponding to wetlands were assigned the value of 0.25, different types of forest and semi-natural areas were assigned the value of 0.5, agricultural areas were assigned 0.75, and artificial surfaces were assigned the value of one.

Finally, regarding the presence of human settlements, we used a modern pan-European dataset obtained from the Copernicus Land Monitoring Service (http://land.copernicus.eu/pan-european/GHSL/european-settlement-map, accessed on 14 September 2021). The human settlements layer is made available at a spatial resolution of 10 m, and it represents the percentage of built-up area coverage per spatial unit, based on SPOT5 and SPOT6 satellite imagery from the year of 2015. The automated information extraction process used for building this dataset uses machine learning techniques in order to understand systematic relations between morphological and textural features, extracted from the multispectral and panchromatic bands of the satellite imagery [38].

### 4.2. Implementation Details

Our disaggregation method, previously described in Section 3, relies on a regression model to combine the ancillary data and produce disaggregation estimates. The entire procedure was implemented in the Python language, using frameworks such as scikit-learn (http://scikit-learn.org, accessed on 14 September 2021) and Tensorflow (http://www.tensorflow.org, accessed on 14 September 2021).

Concerning the regression algorithm, we tested an encoder–decoder CNN and compared its performance against linear regression and random forest algorithms, which process individual cells instead of patches. The first approach consists of a standard linear least-squares fit, which computes a weighted linear combination of the predictive covariates added to a bias term. In turn, a random forest corresponds to an ensemble of decision trees, each corresponding to a non-linear flow-chart-like structure of tests over the values of the attributes. The random forest approach fits different decision trees on random subsets of the instances and features, and averages the results of each tree to improve accuracy and control overfitting [39].

The hyper-parameters of the models were tuned for optimal performance. Regarding the encoder–decoder CNN, we tested loss functions based on the Mean Absolute Error (MAE), the Mean Squared Error (MSE), and the Huber Loss. The best configuration was achieved when the CNN takes input patches of $16 \times 16$ grid cells (i.e., 256 cells in total) and when the training procedure considered batches of 64 instances, using ten epochs over the training data per iteration. We used the Adam optimization algorithm [40] with the default parameters in the Tensorflow library, except for the learning rate, which was set to $10^{-3}$. In the Huber Loss, we used a $\delta$ value of 1. Finally, the value of $w_1$ from Equation (4) was set to 1, $w_2$ ranged from 1 to 1.25, and $w_3$ was set to $10^4$.

### 4.3. Evaluation Metrics

The results were assessed in terms of the Mean Absolute Error (MAE), the Root Mean Square Error (RMSE), and the Coefficient of Determination ($R^2$) between estimated and ground-truth values. The formulas for the MAE and RMSE have already been previously described in Section 3.4, and $R^2$ is defined as follows.

$$R^2(y, \hat{y}) = 1 - \frac{\sum_{i=1}^{n}(y_i - \hat{y}_i)^2}{\sum_{i=1}^{n}(y_i - \bar{y})^2}. \tag{6}$$

In Equation (6), $y_i$ corresponds to a ground-truth value, $\hat{y}_i$ corresponds to a predicted value, $\bar{y}$ is the mean of the ground-truth values, and $n$ is the number of instances. The coefficient of determination $R^2$ measures the proportion of total variation in the ground-truth values that is explained by the model. It specifically assigns a value of one to a model whose predictions exactly match the ground-truth values, a value of zero to a model that always predicts the average, and a negative value to a model that is worse than the baseline corresponding to the average. Different from the MAE and RMSE metrics, in which lower errors indicate better results, higher $R^2$ values correspond to better estimates.

Although these three metrics have advantages and disadvantages, the disaggregation errors computed through a standard MAE are more easily interpretable [41]. In particular, the RMSE between predicted and ground-truth values can over-penalize the outliers. Taking this into account, we attempted to optimize our models for producing lower absolute disaggregation errors, and preferred results with better MAE throughout the article, to the detriment of the remaining metrics.

## 5. Experimental Results

We designed several experiments to evaluate the use of our encoder–decoder CNN within the self-training framework. We first compare the application of the model, with an RMSE loss and without any type of regularization strategy, against baseline disaggregation methods (Section 5.1). In Section 5.2, we test the impact of different initial model estimates, while in Section 5.3 we test stopping criteria for the iterative method. Then, in Section 5.4, we compare different loss functions, and in Section 5.5 we evaluate the strategy to promote equivariance to spatial transformations of the input data. Finally, in Section 5.6, we show the results for different regression models in our self-training framework, while Section 5.7 illustrates the distribution of the disaggregated results and of the standard deviation associated with the model estimates.

The first results (i.e., those presented in Section 5.1) are reported for the variables related to the amount of withdrawals in three different scenarios, i.e., the overall amount for the entire year, the summer period, and the winter period. Intermediate experiments, which evaluate the impact of adding different components into the model (i.e., from Sections 5.2–5.5), are reported only for the variable concerning the overall amount of withdrawals. In Section 5.6, we use the remaining dataset to validate the approach in the disaggregation of another variable, i.e., the number of live births.

All tables with results present the MAE, RMSE, and $R^2$ metrics in two scenarios, namely one corresponding to the values with our stopping criterion, and the other corresponding to the best iteration (among the 30 that were used for self-training). They also report the percentage of the gain obtained over the result achieved when using the baseline corresponding to the smooth weighted interpolation (i.e., the best non-regression baseline in our tests). In order to alleviate problems with random initializations, the values reported for all experiments using CNN models result from averaging two tests. Since the variations between the results of the two experiments were not significant, we considered that it was not necessary to use more than two tests for computing the average. Additionally, values in bold correspond to the best results for each variable.

### 5.1. Proposed Approach

Tables 4–6 show the results obtained when using disaggregation baselines corresponding to (i) mass-preserving areal weighting, (ii) pycnophylactic interpolation, (iii) a weighted interpolation (WI) method which disaggregates the count data proportionally to population distribution, and (iv) the result of applying a smoothing operation over WI, which we named smooth weighted interpolation. Among these, the smooth weighted interpolation achieved the best results for all the variables.

For comparison with the baselines, the tables also present the results obtained with our CNN architecture, trained with a loss function based on the RMSE between predicted and real values. In this scenario, we report the results achieved with and without adding into the CNN architecture the shortcut which directly connects the ancillary variables to the output of the upscaling blocks.

**Table 4.** Results obtained with different disaggregation methods for the overall amount of national withdrawals on automated teller machines.

| | MAE | RMSE | $R^2$ | Gain(%) / Baseline MAE | RMSE | $R^2$ | W/o Stopping Criterion MAE | RMSE | $R^2$ |
|---|---|---|---|---|---|---|---|---|---|
| Areal weighting | 79,667.9 | 205,045.5 | −0.0471 | −364.1 | −218.2 | −105.3 | | | |
| Pycnophylactic interpolation | 78,290.1 | 201,420.9 | −0.0104 | −356.1 | −212.6 | −101.2 | | | |
| Weighted interpolation | 17,237.7 | 64,644.5 | 0.8959 | −0.4 | −0.3 | −0.1 | | | |
| *Smooth weighted interpolation* | *17,166.5* | *64,433.1* | *0.8966* | — | — | — | | | |
| SL w/ CNN (no shortcut) | 15,326.9 | 58,395.4 | 0.9151 | 10.7 | 9.4 | 2.1 | 14,846.7 | 55,228.4 | 0.9240 |
| SL w/ CNN | 15,015.0 | 54,957.0 | 0.9248 | **12.5** | **14.7** | **3.1** | 15,015.0 | 54,957.0 | 0.9248 |

**Table 5.** Results obtained with different disaggregation methods for the amount of national withdrawals during the summer months.

| | MAE | RMSE | $R^2$ | Gain(%) / Baseline MAE | RMSE | $R^2$ | W/o Stopping Criterion MAE | RMSE | $R^2$ |
|---|---|---|---|---|---|---|---|---|---|
| Areal weighting | 26,815.1 | 67,219.3 | −0.0464 | −383.6 | −237.4 | −105.1 | | | |
| Pycnophylactic interpolation | 26,355.0 | 65,990.2 | −0.0085 | −375.3 | −231.2 | −100.9 | | | |
| Weighted interpolation | 5573.6 | 20,001.7 | 0.9073 | −0.5 | −0.4 | −0.1 | | | |
| *Smooth weighted interpolation* | *5545.0* | *19,925.6* | *0.9080* | — | — | — | | | |
| SL w/ CNN (no shortcut) | 5185.8 | 18,514.0 | 0.9206 | 6.5 | 7.1 | 1.4 | 5086.7 | 18,491.5 | 0.9208 |
| SL w/ CNN | 5178.5 | 18,510.5 | 0.9207 | **6.6** | **7.1** | **1.4** | 5081.9 | 18,194.0 | 0.9234 |

**Table 6.** Results obtained with different disaggregation methods for the amount of national withdrawals during the winter months.

| | MAE | RMSE | $R^2$ | Gain(%) / Baseline MAE | RMSE | $R^2$ | W/o Stopping Criterion MAE | RMSE | $R^2$ |
|---|---|---|---|---|---|---|---|---|---|
| Areal weighting | 26,301.8 | 68,711.9 | −0.0471 | −350.2 | −209.1 | −105.3 | | | |
| Pycnophylactic interpolation | 25,840.2 | 67,516.5 | −0.0109 | −342.3 | −203.7 | −101.2 | | | |
| Weighted interpolation | 5865.2 | 22,301.3 | 0.8897 | −0.4 | −0.3 | −0.1 | | | |
| *Smooth weighted interpolation* | *5842.8* | *22,233.1* | *0.8904* | — | — | — | | | |
| SL w/ CNN (no shortcut) | 5178.8 | 20,025.9 | 0.9111 | 11.4 | 9.9 | 2.3 | 5090.7 | 19,800.2 | 0.9131 |
| SL w/ CNN | 5109.8 | 19,624.9 | 0.9146 | **12.5** | **11.7** | **2.7** | 5007.5 | 19,399.2 | 0.9166 |

The results from Tables 4–6 show that the proposed self-training approach outperforms all the non-regression baselines, with improvements of up to 14.7% over the best baseline in terms of the RMSE, and up to 12.5% in terms of the MAE. The incorporation of the shortcut produced better disaggregation estimates (e.g., improvements of 1.8% in MAE and 5.3% in RMSE). It is also interesting to note that, in all the reported experiments, the error values obtained for the iteration corresponding to our stopping criterion (i.e., the first three columns of the tables) are close to the best possible error values overall (i.e., the last three columns of the tables).

### 5.2. Initial Estimates

Figure 6 details the results of the self-training procedure with one of the tested variables, specifically the one corresponding to the overall amount of withdrawals, during 30 iterations. This figure illustrates the importance of choosing a good heuristic for producing the initial estimates, as discussed in Section 3.1. It compares, side-by-side, the application of the proposed method when leveraging three heuristics for producing the initial estimates, namely (i) the pycnophylactic interpolation, (ii) the weighted interpolation, and (iii) the smooth weighted interpolation. In each figure, we show the behavior of our disaggregation method, as well as the result of the corresponding heuristic used for the initial estimates.

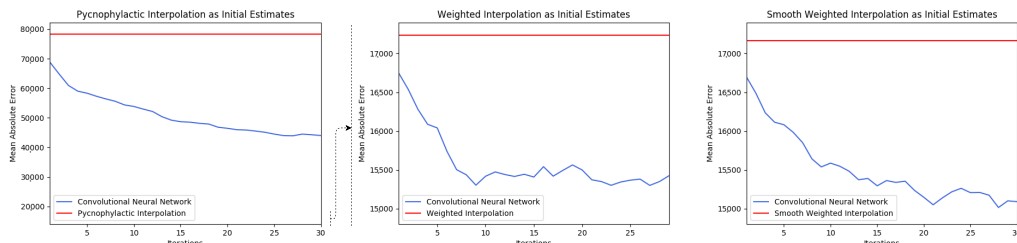

**Figure 6.** Error values when using different heuristics for producing the initial estimates in the proposed self-training approach for the overall amount of national withdrawals.

From Figure 6, we can see that the model achieves much better results when prior useful information is provided as initial estimates, such as in the case of both weighted interpolation and smooth weighted interpolation. This is an expected result, since the initialization corresponding to the pycnophylactic interpolation consists of very smooth and uniform estimates, and is therefore much worse in terms of providing interesting data patterns that could be learned. The use of the CNN when leveraging more useful information specifically results in around three times fewer disaggregation errors when compared to using worse initial estimates.

From the figure, we can also infer that our method benefits from the execution of several iterations, since better disaggregation results are always obtained after Iteration 1. The advantages of the iterative self-training process are particularly noticeable when the initial estimates contain less useful information (e.g., as in the case of the pycnophylactic interpolation), in which the CNN achieves improvements of about 35% when compared to the first iteration. When initial estimates with more useful information are used, although

the result obtained in Iteration 1 is much better when compared to worse initial estimates, the improvements achieved with multiple iterations are of about 10%.

### 5.3. Stopping Criterion

The stopping criterion of the algorithm is also an important part of our approach. As discussed in Section 3.4, we empirically concluded that the standard deviation of the results at each iteration is a good proxy for the quality of the disaggregation results at that iteration. Figure 7 illustrates this conclusion by comparing this metric against an alternative that involved using the original source zones (i.e., the polygons from which we have the original aggregated data) to infer the quality of the results which will be produced in the final target zones. To this end, we first created artificial pairs of source zones by merging neighbor polygons and by summing the associated counts. Then, at each iteration, we computed the errors associated with re-aggregating the estimated disaggregation results, computed when using the artificial pairs of polygons as source zones, at the level of the original source polygons. We specifically plot, side-by-side, (i) the MAE at each iteration when using our method, (ii) the standard deviation of the results at the same iteration, and (iii) the $R^2$ between the known values at each source zone against estimates resulting from disaggregating the counts when using artificial pairs of polygons as source zones. All the values are compared at the same scale by performing a min–max normalization. In the case of the standard deviation and the $R^2$ metrics, we present the complement of the normalized values.

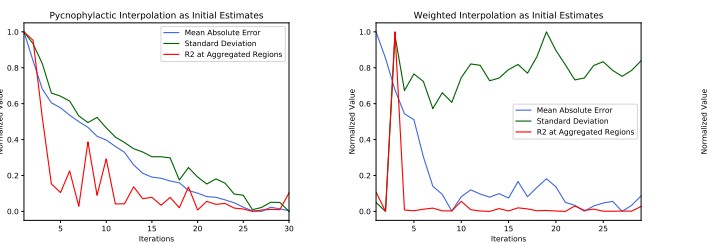
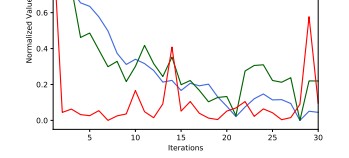

**Figure 7.** Disaggregation errors together with the standard deviation of the results for the overall amount of national withdrawals.

We can verify the potential of this stopping criterion in three aspects: first, the speed of its decrease over the 30 iterations follows the behavior of the MAE closely. Then, as illustrated in the rightmost figure which corresponds to the experiment that obtained the best result, the minimum standard deviation value over the 30 iterations often corresponds to the minimum MAE value. We can conclude that, through the proposed stopping criterion, it is possible to detect which iterations are most likely to correspond to the best MAEs, avoiding the need to execute the 30 iterations.

Different from the behavior of the standard deviation metric, the $R^2$ coefficient has a rapid decrease in the first iterations. This may be because the task measured by $R^2$, i.e., disaggregating from artificial pairs of source zones to the original source polygons, is much easier than the original task of disaggregating from source to target zones. In fact, while in each artificial pair of source zones there are only two source polygons contained within, we have that in each source zone there are never fewer than 10 target zones (i.e., the task is harder, given the much smaller target regions). Given this, we used the standard deviation metric as the stopping criterion on the remaining experiments.

### 5.4. Loss Functions

Table 7 details the disaggregation results for the variable corresponding to the overall number of withdrawals. It specifically reports the results when testing the six loss functions discussed in Section 3, namely (i) the MAE loss, (ii) the MAE with a penalty corresponding to the standard deviation of the produced map, (iii) the RMSE loss, (iv) the RMSE with the

standard deviation penalty, (v) the Huber Loss, and (vi) the Huber Loss with the standard deviation penalty.

**Table 7.** Results obtained with the application of the encoder–decoder model leveraging different losses for the overall amount of national withdrawals.

| | MAE | RMSE | $R^2$ | Gain(%) / Baseline MAE | RMSE | $R^2$ | W/o Stopping Criterion MAE | RMSE | $R^2$ |
|---|---|---|---|---|---|---|---|---|---|
| MAE Loss | 14,423.2 | 57,019.1 | 0.9190 | 16.0 | 11.5 | 2.5 | 14,395.4 | 57,074.9 | 0.9189 |
| MAE Loss w/Penalty | 14,726.3 | 57,646.5 | 0.9172 | 14.2 | 10.5 | 2.3 | 14,508.3 | 57,366.0 | 0.9180 |
| RMSE Loss | 15,015.0 | 54,957.0 | 0.9248 | 12.5 | 14.7 | 3.1 | 15,015.0 | 54,957.0 | 0.9248 |
| RMSE Loss w/Penalty | 14,981.6 | 55,792.0 | 0.9225 | 12.7 | 13.4 | 2.9 | 14,981.6 | 55,792.0 | 0.9225 |
| Huber Loss | 14,314.4 | 57,241.5 | 0.9184 | 16.6 | 11.2 | 2.4 | 14,314.4 | 57,241.5 | 0.9184 |
| Huber Loss w/Penalty | 14,282.6 | 57,068.0 | 0.9189 | **16.8** | 11.4 | 2.5 | 14,240.9 | 56,766.7 | 0.9197 |

Table 7 shows that the best results for the RMSE and $R^2$ metrics were obtained with the RMSE loss between predicted and true patches. However, the lowest MAE errors were achieved when using the Huber Loss, which also shows, as expected, a trade-off between the RMSE and the MAE metrics. In this specific scenario, i.e., when obtaining the best MAE values through using a Huber Loss, one can also observe that it is helpful to use the standard deviation for improving all the error metrics. Taking all these elements into account, and given that the Huber Loss is a good compromise between the strengths and weaknesses of both the MAE and RMSE losses, we used it in conjunction with the standard deviation penalty as the loss function for the remaining experiments. Similar to what happened in the previous tables, the standard deviation of the produced map remains a good proxy for the best iteration of the model. In particular, in three of the six reported experiments, the results obtained when using this metric as a selection criterion, and the best results, are coincident.

*5.5. Equivariance to Transformations*

Following the discussion from Section 3.4, we also tested the incorporation of equivariance to spatial transformations into the network. One can expect that, with this new supervision signal, the convergence behavior of the model suffers fewer oscillations. As a consequence of this smoother process, we can also have better disaggregation results. In Table 8, we compare this approach with the application of data augmentation external to the model, using the same transformations. The purpose of this experiment was to validate that a potential improvement would not only result from having more examples to learn, but also from the way the network uses the relations between the examples in order to improve generalization. From the results of Table 8, we can infer the benefit of this strategy, which produced better MAE values, in comparison to the same model without promoting equivariance and data augmentation.

**Table 8.** Results obtained when applying equivariance to transformations over the model for the overall amount of national withdrawals.

| | MAE | RMSE | $R^2$ | Gain(%) / Baseline MAE | RMSE | $R^2$ | W/o Stopping Criterion MAE | RMSE | $R^2$ |
|---|---|---|---|---|---|---|---|---|---|
| CNN | 14,282.6 | 57,068.0 | 0.9189 | 16.8 | 11.4 | 2.5 | 14,240.9 | 56,766.7 | 0.9197 |
| CNN w/Augmentation | 14,591.6 | 59,228.0 | 0.9126 | 15.0 | 8.1 | 1.8 | 14,591.6 | 59,228.0 | 0.9126 |
| CNN w/Equivariance | 14,159.7 | 57,807.6 | 0.9167 | **17.5** | 10.3 | 2.2 | 14,152.4 | 57,489.9 | 0.9177 |

Figure 8 compares the results obtained in the experiment that incorporates equivariance to transformations of the input data (i.e., when using a Huber Loss and all the regularization strategies) against the ones obtained with a RMSE loss and without regularization (i.e., reported in the last row of Table 4). In both cases, the figure presents the evolution of the disaggregation errors over the 30 iterations together with the metric

used as stopping criterion (i.e., the standard deviation of the produced patches). From the figure, we conclude that the use of the Huber Loss with different regularization strategies produced a smoother and faster convergence process, which is closely linked to the high-quality disaggregated values that are obtained in this test.

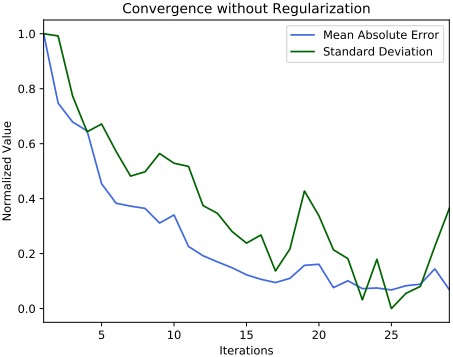 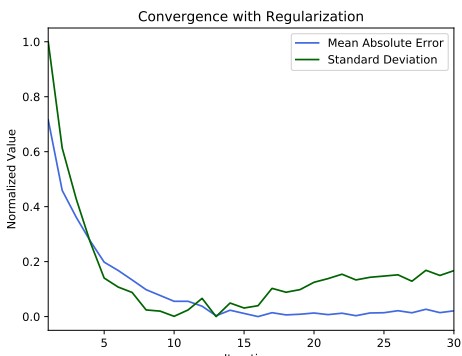

**Figure 8.** Disaggregation errors together with the standard deviation of the results, with and without using regularization strategies, for the overall amount of national withdrawals.

### 5.6. Different Regression Algorithms and Variables

In Tables 9–11, we summarize results for the three datasets used in the tests so far and compare them with the application of the self-training approach using alternative regression models. In particular, we report, for each case, the disaggregation results with (i) the best baseline, i.e., smooth weighted interpolation, (ii) the result of replacing the regression algorithm of the proposed approach with linear regression, (iii) the result of replacing the regression algorithm of the proposed approach with a random forest model, (iv) the proposed approach using an RMSE loss and without incorporating regularization strategies, and (v) the proposed approach with regularization using a Huber Loss. From these tables, one can confirm the good performance of the proposed method. The optimized version produces improvements over the best non-regression baseline of up to 19.4% in terms of the MAE, which also represents a considerable increase when compared to using the same model without the tested strategies (i.e., an increase from 12.5% to 19.4%). Our approach also produced better disaggregation results, when compared to the application of the self-training framework with alternative regression models, in the majority of the experiments. In particular, it outperformed the linear regression in all scenarios and the random forest model in two of the three tested variables. In the remaining variable, it achieved similar results.

**Table 9.** Results obtained with the self-training approach using different regression algorithms for the overall amount of national withdrawals.

| | MAE | RMSE | $R^2$ | Gain(%) / Baseline | | | W/o Stopping Criterion | | |
| | | | | MAE | RMSE | $R^2$ | MAE | RMSE | $R^2$ |
| --- | --- | --- | --- | --- | --- | --- | --- | --- | --- |
| *Smooth weighted interpolation* | 17,166.5 | 64,433.1 | 0.8966 | — | — | — | | | |
| SL w/linear model | 17,083.9 | 61,438.7 | 0.9060 | 0.5 | 4.6 | 1.0 | 16,826.0 | 59,999.8 | 0.9103 |
| SL w/random forests | 14,582.4 | 56,598.0 | 0.9202 | 15.1 | 12.2 | 2.6 | 14,582.4 | 56,598.0 | 0.9202 |
| SL w/baseline CNN | 15,015.0 | 54,957.0 | 0.9248 | 12.5 | **14.7** | **3.1** | 15,015.0 | 54,957.0 | 0.9248 |
| SL w/optimized CNN | 14,159.7 | 57,807.6 | 0.9167 | **17.5** | 10.3 | 2.2 | 14,152.4 | 57,489.9 | 0.9177 |

**Table 10.** Results obtained with the self-training approach using different regression algorithms for the amount of national withdrawals during the summer months.

| | MAE | RMSE | R² | Gain(%) / Baseline MAE | Gain(%) / Baseline RMSE | Gain(%) / Baseline R² | W/o Stopping Criterion MAE | W/o Stopping Criterion RMSE | W/o Stopping Criterion R² |
|---|---|---|---|---|---|---|---|---|---|
| *Smooth weighted interpolation* | 5545.0 | 19,925.6 | 0.9080 | — | — | — | | | |
| SL w/linear model | 5777.6 | 18,961.7 | 0.9167 | −4.2 | 4.8 | 1.0 | 5727.7 | 19,152.4 | 0.9150 |
| SL w/random forests | 4929.6 | 17,574.0 | 0.9285 | 11.1 | **11.8** | **2.3** | 4929.6 | 17,574.0 | 0.9285 |
| SL w/baseline CNN | 5178.5 | 18,510.5 | 0.9207 | 6.6 | 7.1 | 1.4 | 5081.9 | 18,194.0 | 0.9234 |
| SL w/optimized CNN | 4902.7 | 18,569.9 | 0.9202 | **11.6** | 6.8 | 1.3 | 4895.5 | 18,583.5 | 0.9200 |

**Table 11.** Results obtained with the self-training approach using different regression algorithms for the amount of national withdrawals during the winter months.

| | MAE | RMSE | R² | Gain(%) / Baseline MAE | Gain(%) / Baseline RMSE | Gain(%) / Baseline R² | W/o Stopping Criterion MAE | W/o Stopping Criterion RMSE | W/o Stopping Criterion R² |
|---|---|---|---|---|---|---|---|---|---|
| *Smooth weighted interpolation* | 5842.8 | 22,233.1 | 0.8904 | — | — | — | | | |
| SL w/linear model | 5746.4 | 21,169.5 | 0.9006 | 1.6 | 4.8 | 1.1 | 5588.3 | 20,508.4 | 0.9067 |
| SL w/random forests | 4919.2 | 19,569.2 | 0.9151 | 15.8 | 12.0 | 2.8 | 4919.2 | 19,569.2 | 0.9151 |
| SL w/baseline CNN | 5109.8 | 19,624.9 | 0.9146 | 12.5 | 11.7 | 2.7 | 5007.5 | 19,399.2 | 0.9166 |
| SL w/optimized CNN | 4707.6 | 19,235.0 | 0.9179 | **19.4** | **13.5** | **3.1** | 4703.0 | 19,401.0 | 0.9165 |

Figure 9 compares the behavior of the self-training framework for three regression algorithms: our optimized encoder–decoder CNN, a linear regression, and a random forest model. It specifically plots, side-by-side, the disaggregation results during 30 iterations of the algorithm, obtained for the variables corresponding to (i) the overall amount of withdrawals, (ii) the amount of withdrawals during the summer, and (iii) the amount of withdrawals during the winter. From the figure, we can see that the use of alternative algorithms tends to result in a higher MAE. More specifically, the CNN has a better MAE than the alternative models over almost all the 30 iterations, and in particular in the first iteration. This demonstrates that even without the application of a self-training framework (i.e., if we only ran the algorithms for one iteration), the use of our encoder–decoder CNN could lead to better results. Among the different regression algorithms, the random forest appears to benefit more from the execution of several iterations, which allows it to have significantly better overall results than the linear regression, despite not always having the best result in the first iteration. Second, the results highlight once again the importance of choosing appropriate stopping criteria, since in the case of the CNN and the linear regression models, better results are obtained before iteration 30.

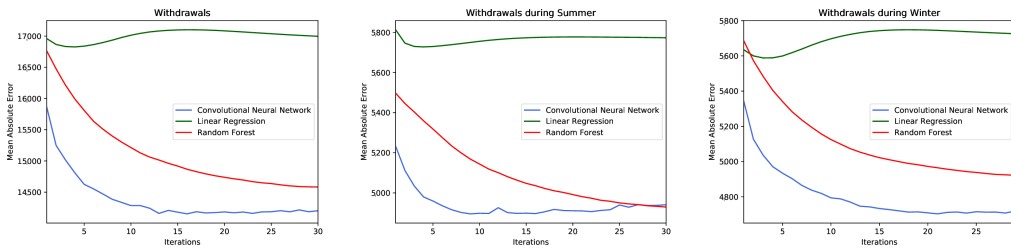

**Figure 9.** Error values when using different regression algorithms in the proposed self-training approach for the overall amount of national withdrawals, the amount of national withdrawals during the summer, and the amount of national withdrawals during the winter.

In Table 12, we report the results obtained from further validating the disaggregation method in a separate variable, i.e., the number of live births. We specifically compare the results obtained when leveraging (i) the same baselines presented in previous tables, and (ii) the proposed approach when leveraging a Huber Loss and all the regularization strategies that were tested. From this table, one can confirm again the potential of our method, since it achieves the best results in all metrics (e.g., improvements of 12% in terms of the MAE over the best non-regression baseline).

**Table 12.** Results obtained with the self-training approach using different regression algorithms for the number of live births.

| | MAE | RMSE | R² | Gain(%) / Baseline | | | W/o Stopping Criterion | | |
| --- | --- | --- | --- | --- | --- | --- | --- | --- | --- |
| | | | | MAE | RMSE | R² | MAE | RMSE | R² |
| Areal weighting | 20.9 | 59.9 | 0.3354 | −318.0 | −403.4 | −65.6 | | | |
| Pycnophylactic interpolation | 19.4 | 53.9 | 0.4615 | −288.0 | −352.9 | −52.6 | | | |
| Weighted interpolation | 5.0 | 11.4 | 0.9759 | 0.0 | 4.2 | 0.2 | | | |
| *Smooth weighted interpolation* | *5.0* | *11.9* | *0.9740* | — | — | — | | | |
| SL w/linear model | 5.0 | 11.3 | 0.9763 | 0.0 | 5.0 | 0.2 | 4.8 | 11.4 | 0.9758 |
| SL w/random forests | 4.5 | 11.2 | 0.9766 | 10.0 | 5.9 | 0.3 | 4.4 | 10.8 | 0.9785 |
| SL w/optimized CNN | 4.4 | 10.7 | 0.9788 | **12.0** | **10.1** | **0.5** | 4.4 | 10.7 | 0.9788 |

*5.7. Qualitative Analysis*

In Figure 10, we present the resulting disaggregation map for the variable corresponding to the overall amount of withdrawals when using as source zones the lower aggregation units available (i.e., municipalities). We specifically plot for the territory of Continental Portugal, side-by-side, (i) the map obtained with the baseline disaggregation method corresponding to the smooth weighted interpolation, (ii) a map obtained when using a linear model as the regression algorithm within the self-training framework, and (iii) a map obtained when using the proposed encoder–decoder CNN with a Huber Loss and all the regularization strategies. For illustration purposes, we also present a zoom on the regions corresponding to the South Alentejo and the Algarve for the three methods. In general, one can see that more-developed areas have higher values for the withdrawals. Moreover, the baseline corresponding to smooth weighted interpolation, as well as the application of the self-training approach with linear regression, produced interesting estimates. However, both approaches tend to result in insufficient detail in some areas (e.g., in regions in the south of Portugal, such as the Alentejo). Oppositely, the CNN produced results with higher spatial detail in these regions.

In Figure 11, we assess the uncertainty of the produced estimates by showing the respective standard deviation. We take advantage of the fact that the disaggregated values for each cell are calculated by averaging all the corresponding outputs from overlapping patches. We plot a grid which shows (i) a map containing the standard deviation associated with averaging the different pixel values from the different patches that contain them (at the right), (ii) a scatter-plot that compares the standard deviation values against the produced disaggregated results at the level of 200 × 200 m cells (bottom left), and (iii) a violin plot that contrasts the standard deviation values with the error that is subsequently obtained at the level of the corresponding municipalities by showing the associated distribution in municipalities with error values lower or higher than the median error. In this figure, all the values result from disaggregating the variable corresponding to the overall number of withdrawals at the NUTS III level. From the scatter-plot and from the map, we can conclude that higher disaggregated counts are also associated with higher uncertainty in the production of the results. One can also notice from the violin plot that estimation uncertainties are also correlated to zones that end up obtaining higher disaggregation errors. The distribution of the standard deviation in municipalities with disaggregation errors above the median (i.e., the right part of the chart) is wider, and includes the highest standard deviation values observed, when compared to the case of municipalities with disaggregation errors lower than the median (i.e., the left part of the chart).

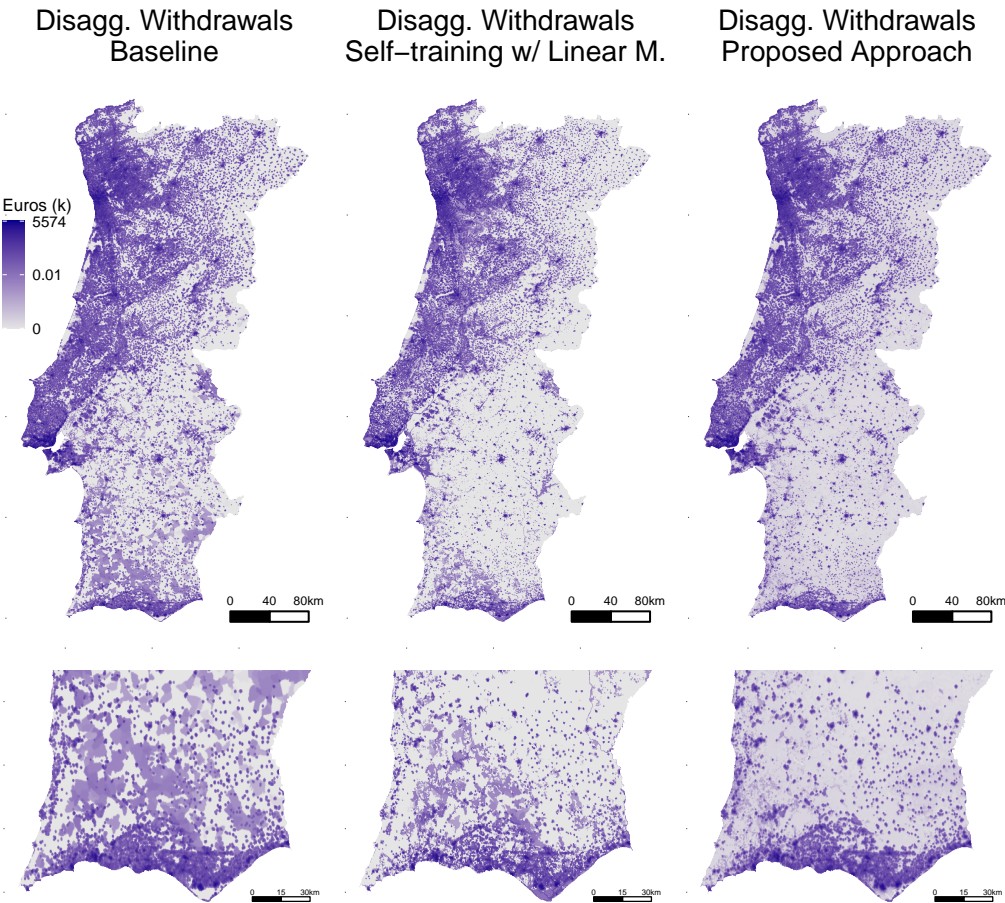

**Figure 10.** Spatially disaggregated results for the variable corresponding to the overall number of national withdrawals.

Figure 12 details the variation between the disaggregated results corresponding to different periods of the year. We attempted to check for patterns associated with the well-known movement of people in the summer and winter period in areas related to tourism such as the Algarve (i.e., in the south of the country). For this same region, we present (i) the monthly average of the overall number of withdrawals, (ii) the difference between the monthly average of the overall number of withdrawals against the monthly average during the summer period, (iii) the difference between the monthly average of the overall number of withdrawals against the monthly average during the winter period, and (iv) the difference between the monthly average of the number of withdrawals during summer against the monthly average during the winter. All the results included in this figure were obtained when leveraging our approach with a Huber Loss and including all the regularization strategies and when disaggregating data collected at the level of municipalities. The figure shows, as expected, a higher volume of withdrawals in coastal regions (e.g., corresponding to regions known to be tourist attractions, such as Portimão or Albufeira) during the summer.

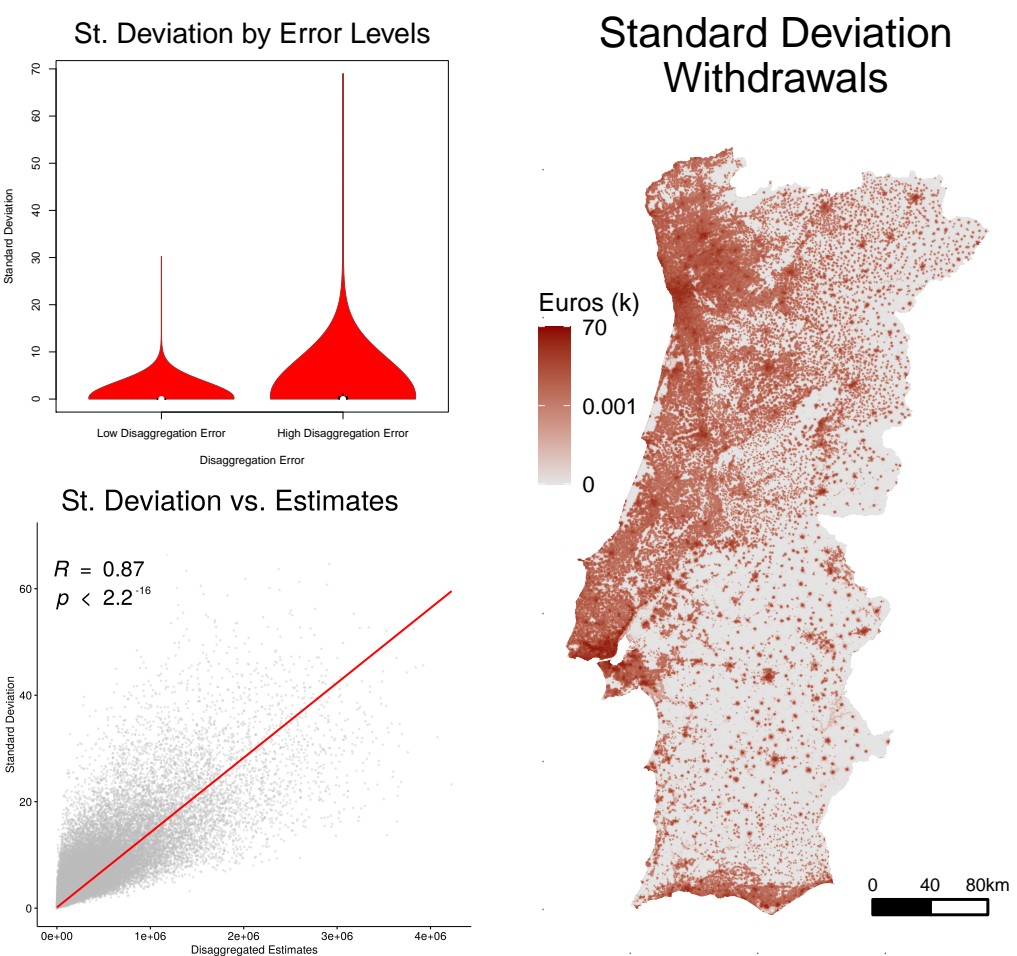

**Figure 11.** Standard deviation associated with the disaggregated raster cells for the variable corresponding to the overall number of national withdrawals.

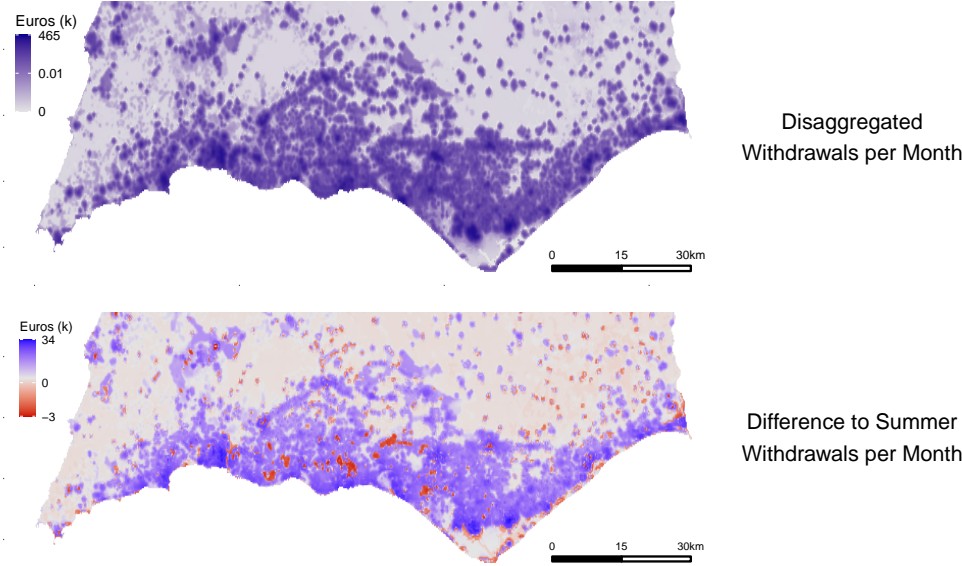

**Figure 12.** *Cont.*

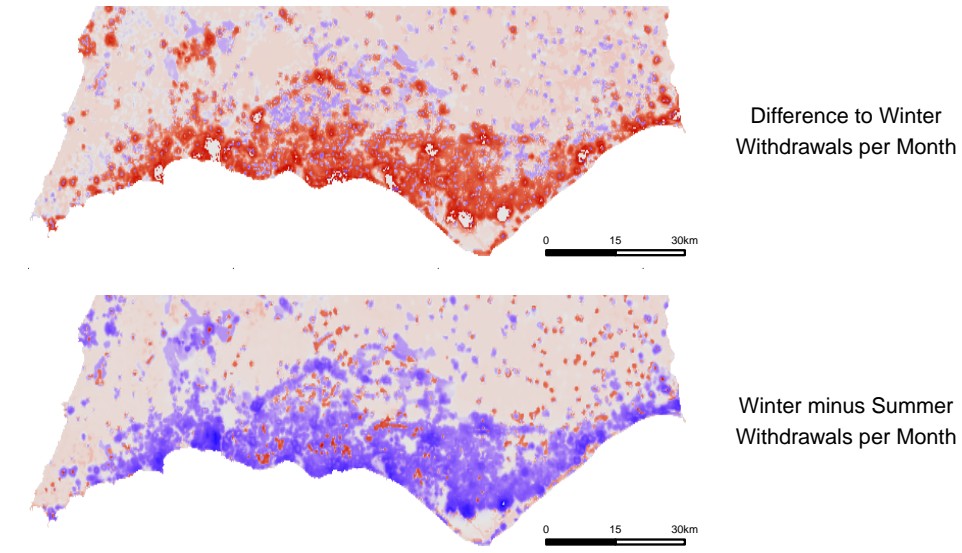

**Figure 12.** Difference between the disaggregated results for the variables corresponding to the number of withdrawals during summer and the number of withdrawals during winter.

## 6. Conclusions and Future Work

This article reported on a novel spatial disaggregation method that uses self-training. We specifically used an encoder–decoder CNN for iteratively refining initial estimates produced by a seminal disaggregation heuristic, such as pycnophylactic interpolation. We presented results that indicate that the proposed method outperforms seminal disaggregation baselines, or variations of the proposed approach that use more traditional regression models (i.e., linear regression or random forests, which have been commonly used in previous studies). We also showed how successive iterations of our self-training approach improve disaggregation results over the use of a single iteration with any of the considered regression models, particularly when considering our CNN guided by tailored regularization strategies.

Our method was evaluated on variables collected for Continental Portugal (i.e., the number of withdrawals on automated teller machines and the number of live births). However, it can naturally also be applied to the disaggregation of other types of socio-economic variables concerning other territories. The proposed method can thus have many practical applications. For instance, disaggregated socio-economic statistical data can inform the quantitative assessment and the dynamic monitoring of the sustainable development goals from the United Nations 2030 agenda [8]. We effectively combined different ideas from the literature (see Table 1 for a summary), improving on previous methods through the combination of deep learning methods with self-training for spatial data disaggregation. Besides disaggregation, we believe that similar ideas can also have applications in other ill-posed and/or generative spatial modeling tasks, such as data interpolation [42,43].

Looking closer at the obtained results, we can see that seminal disaggregation methods (i.e., mass-preserving areal weighting or pycnophylactic interpolation) are 300–400 times worse than a non-regression baseline that uses the population distribution. An adapted version of this method (i.e., smooth weighted interpolation) was used to initialize the self-training procedure. In doing so, our method achieved improvements of up to 14.7% in RMSE, and up to 12.5% in MAE, over the stronger non-regression baseline. We also saw further improvements when optimizing the CNN model, used within the self-training procedure, in order to (i) better use the input data associated to each individual raster cell, (ii) be less susceptible to strong outliers, (iii) force heterogeneity in the disaggregated results, and (iv) promote equivariance to spatial transformations. The percentage of gains

can increase from 12.5% in MAE to 19.4%, and all the tested strategies had an impact on improving the error values (e.g., gains of 1.8% when adding a shortcut connection into the network, 0.2% when promoting heterogeneity in the results, or 0.7% when encoding equivariance to spatial transformations).

Despite the interesting results, there are also many open challenges for future work. The general method reported in this article can, for instance, easily be extended to consider different regression algorithms or combinations of algorithms. Different CNN or transformer-based architectures can be used in place of the U-Net model, similar to those that are currently achieving state-of-the-art results on different types of image processing and computer vision problems (e.g., in tasks such as segmentation or image super-resolution). Moreover, our experimental results showed that the application of the self-training framework with regression algorithms such as random forests is also competitive. Taking this into account, it may be interesting to investigate if co-training strategies, in which models using individual pixels are trained alternately with an encoder–decoder CNN, can be used to improve results. In addition, our approach can be extended to explore other sources of ancillary data as model features (e.g., information on points-of-interest [12,13]), other loss functions (e.g., using Tukey's biweight loss function as an alternative to the Huber Loss [44] or using additional components in the global loss), or auxiliary prediction tasks. Concerning the use of additional loss components, one possibility includes comparing patches computed in iterations far away from each other within the self-training procedure. We expect the disaggregation results to improve over the self-training iterations, and thus the values produced in later iterations should be different from those produced initially. Concerning the use of auxiliary prediction tasks, we can consider extending the encoder–decoder model for multi-task learning, producing other related predictions in addition to the disaggregated values. Examples include predicting land usage information, when available for model training, or a local spatial autocorrelation index [43]). Auxiliary prediction tasks can perhaps help the model to learn interesting local spatial properties, complementing the learning of the primary task.

It is also important to notice that we rely on a sliding window approach to process/generate patches of $16 \times 16$ cells with the target predictions and the estimates regarding the variation of the results. The stride of the sliding window defines the size of the overlapping regions between two consecutive patches, and consequently the number of predictions that are considered for averaging the results. Most self-training approaches adopt confidence measures to select and/or weight the instances for the next iteration, on the basis of the confidence over results produced by previous iterations. In future work, the same procedure used for computing the variation in the predicted results can be used to weight the input patches with the ancillary variables, so that the training procedure assigns higher importance to those instances likely to be associated with lesser errors. Alternative strategies may also include using more robust statistics to combine the patches, such as the median of the different values instead of the average.

Furthermore, in terms of future work, we plan to extend our disaggregation methodology by leveraging an adversarial learning framework [42,43]. The encoder–decoder model can be seen as a generator, and one can simultaneously train an auxiliary model (i.e., a discriminator) to differentiate between more realistic and less plausible patches. The results of this discriminator can be used as yet another component in the complete loss function. However, one challenge in implementing this idea again relates to the difficulty in obtaining ground-truth data (i.e., real patches of disaggregated data supporting the discrimination between instances). Our particular data disaggregation application would thus be different from the standard training of a generative adversarial network.

**Author Contributions:** Conceptualization, João Monteiro, Bruno Martins, Miguel Costa, and João M. Pires; Methodology, João Monteiro, Bruno Martins, Miguel Costa, and João M. Pires; Software, João Monteiro; Validation, João Monteiro; Formal Analysis, João Monteiro, Bruno Martins, Miguel Costa, and João M. Pires; Investigation, João Monteiro, Bruno Martins, Miguel Costa, and João M. Pires; Resources, João Monteiro and Bruno Martins; Data Curation, João Monteiro; Writing—Original Draft Preparation, João Monteiro; Writing—Review & Editing, João Monteiro, Bruno Martins, Miguel Costa, and João M. Pires; Visualization,João Monteiro and João M. Pires; Supervision, Bruno Martins, Miguel Costa, and João M. Pires. All authors have read and agreed to the published version of the manuscript.

**Funding:** This work was partially supported by Thales Portugal, through the Ph.D. scholarship of João Monteiro, and also by national funds through Fundação para a Ciência e Tecnologia (FCT), under the MIMU project with reference PTDC/CCI-CIF/32607/2017, and also under the INESC-ID multi-annual funding from the PIDDAC program (UIDB/50021/2020).

**Data Availability Statement:** Not Applicable.

**Acknowledgments:** We gratefully acknowledge the support of NVIDIA Corporation, with the donation of the two Titan Xp GPUs used in our experiments.

**Conflicts of Interest:** The authors declare no conflict of interest. The funders also had no role in the design of the study; in the collection, analyses, or interpretation of data; in the writing of the manuscript; or in the decision to publish the results.

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
