# Peer review of "Geospatial Data Disaggregation through Self-Trained Encoder–Decoder Convolutional Models"

_ijgi, doi:10.3390/ijgi10090619_

Round 1
Reviewer 1 Report
The authors use an encoder-decoder convolutional neural network similar to those adopted in studies processing remote-sensing data for land coverage classification and/or image segmentation [15]. Then you should show the differences from [15]? Should make a comparison to indicate how you improve the performance over [15].
In terms of self-training, should use the examples studied in this paper to compare with the results in [16]. What have you done to make improvement over the work in [16]?
The authors explored the use of a weighted surface with values that are proportional to the population density, for disaggregating other variables. More recent methods leverage regression analysis and machine learning to automatically learn the dissymmetric weights [4,6,18,19]. How do you compare your results with those in [4,6,18,19]?
There are several other related studies mentioned in this paper.
We expect to see more comparisons and better descriptions on how this paper advances the state of the art techniques over these studies.
Author Response
Many thanks for the comments. We have addressed the following points:
Point 1: The authors use an encoder-decoder convolutional neural network similar to those adopted in studies processing remote-sensing data for land coverage classification and/or image segmentation [15]. Then you should show the differences from [15]? Should make a comparison to indicate how you improve the performance over [15].
Response 1: In our study, which aimed to spatial disaggregate count data, we used a CNN architecture (i.e., the U-Net) initially proposed for another task – segmenting biomedical images. More recent studies have used the same architecture in spatial tasks, such as the work from Jacobs et al., which was also surveyed. These studies have adapted the original architecture for these different scenarios, for instance, by replacing the model loss into one suitable for regression tasks, instead of those used in classification tasks. In our study, we have also incorporated, into the architecture, a shortcut that directly connects the ancillary data to the last layer of the model. Despite these differences, which make it impossible to directly compare our work to the original work from Ronneberger et al., we present in Tables 4, 5, and 6, the result of using our architecture without the more significant modifications, i.e., using a RMSE loss without the shortcut connection. To the best of our knowledge, this is the closest comparison that we can make to the study from Ronneberger et al., given the differences between tasks and data.
Point 2: In terms of self-training, should use the examples studied in this paper to compare with the results in [16]. What have you done to make improvement over the work in [16]?
Response 2: The self-training methodology which was used in our study consists in two phases. First, we use a spatial disaggregation heuristic for translating the original count data from the source zones to the target resolution. Then, we iteratively refine these estimates, by applying a regression model over the results computed using the disaggregation heuristic, through the efficient combination of a set of ancillary data. Despite the different tasks that are involved, this methodology is similar to the teacher/student framework outlined by Xie et al., which was originally proposed for image classification. In our work, the mentioned teacher/student approach is mapped into the spatial disaggregation scenario, by considering the disaggregation heuristic from phase 1 as the teacher, and the regression algorithm from phase 2 as the student. Apart from this, we borrowed key ideas from the study from Xie et al., who stated that incorporating model regularization in such self-training techniques is crucial. Therefore, and besides the dropout regularization which is used within the U-Net architecture, we also used losses that leverage penalties based on the standard deviation of the produced results, as well as on results computed from transformed versions of the input data. To highlight these main differences, we revised the introduction of the paper accordingly.
Point 3: The authors explored the use of a weighted surface with values that are proportional to the population density, for disaggregating other variables. More recent methods leverage regression analysis and machine learning to automatically learn the dissymmetric weights [4,6,18,19]. How do you compare your results with those in [4,6,18,19]?
Response 3: Some of these studies, such as the ones from Lin et al. and Goerlich et al., took advantage of the fact that land cover is a good proxy for inferring the distribution of population, and used it as a unique source of ancillary data in the disaggregation procedure. Goerlich et al. specifically claimed that the relationship between population and land cover exhibited spatial nonstationarity, and used it in a geographically weighted regression algorithm for areal interpolation. More recent studies, such as the ones from Monteiro et al. and Stevens et al., leveraged multiple sources of ancillary data (i.e., land coverage, together with other products such as night-time lights, or distance from water bodies) to aid in inferring the distribution of different socio-economic indicators. While Stevens et al. leveraged random forests for combining these different data, Monteiro et al. tested the application of different regression models, ranging from linear regression to ensembles of decision trees, in a self-learning framework. Despite the difficulty in comparing our method with the mentioned studies due to the differences in the data that is being downscaled, we argue that our model is comparable to the one from Stevens et al. when we present the disaggregation results leveraging random forest regression, and when only using one iteration of our self-training framework. We compared this alternative against the results when using other regression models, with and without multiple iterations – Figure 9. We also argue that our method is comparable to the one from Monteiro et al. when we use regression models similar to those used in the mentioned study, and compare their application against our regression model (i.e., a U-Net architecture) – Tables 9, 10, 11 and 12.
Point 4: We expect to see more comparisons and better descriptions on how this paper advances the state of the art techniques over these studies. There are several other related studies mentioned in this paper.
Response 4: To highlight the main differences between our study and the ones from the literature, we revised the introduction and the conclusion of the paper accordingly. We also added a table at the end of the Related Work section, which summarizes and highlights the key differences between all the studies that were surveyed in the corresponding section, and compares them to our work. Our approach combines different ideas from the literature, some of them used in tasks and data which were different from our scenario, and uses them for the spatial disaggregation of socio-economic indicators related to the Portuguese territory. We specifically (i) leveraged regression analysis for combining ancillary data and derive the dasymetric weights, similar to Stevens et al. and Cheng et al., (ii) relied on self-training to avoid the need for ground-truth data available at the target resolutions, such as Malone et al. and Monteiro et al., and (iii) used deep learning methods as regression algorithms, such as Tiecke et al., Robinson et al., and Jacobs et al. Apart from the combination of these innovative ideas, and different from the works that were mentioned, we tackle the more challenging task of spatial disaggregating different socio-economic indicators, instead of just population density. Also, we incorporated deep learning models into self-training frameworks. This avoided the need for high-resolution data for model training, but implied additional efforts in terms of model regularization, selecting appropriate loss functions, or choosing valid stopping criteria. For each study in the table, present a column that illustrates the task which was addressed in the corresponding study. We argue that the comparison between the results of our study and the ones from the literature is not fully feasible, due to the differences between tasks and data used. However, we present, throughout the article, results that try to replicate what was carried out in those works, namely by using similar frameworks, such as the removal of the self-training component (i.e., by using only one iteration of the algorithm), or the application of the regression models that were tested in such studies.
Best regards,
João Monteiro
Reviewer 2 Report
1. In the model used in this article, the input patches are directly fused with the data of the last module, which may weaken the model's ability to learn samples. Please analyze the reasons for this improvement in more depth.
2. In the Downscaling block module, why is there an activation function after Max Pooling? Please explain why.
3. In theory, a reasonable loss function should be presented and an explanation of why these functions are used is needed.
4. The figure of structure of training deep convolutional neural networks are needed to show for some interested readers.
Author Response
Many thanks for the comments. We have addressed the following points:
Point 1: In the model used in this article, the input patches are directly fused with the data of the last module, which may weaken the model's ability to learn samples. Please analyze the reasons for this improvement in more depth.
Response 1: The use of shortcut connections was introduced by He et al. (1), who concluded that their incorporation leads to neural networks easier to optimize, at the same time producing improved model accuracy when depths are increased. Our intuition for adding the mentioned connection in our network relied on these conclusions, as well as on the fact that the information from the different variables, provided as ancillary data for each individual cell, is rich enough to be used by itself, i.e., without any type of spatial processing. By adding the shortcut between the input patches and the last module of the U-Net, the model is then able to choose between the direct use of the rich channel information contained in the ancillary data per cell, or the result of taking the associated spatial context into account, through the application of successive convolutions on the data. The mentioned idea was validated in Tables 4, 5, and 6, by comparing the use of the U-Net architecture with and without the shortcut connection, and by concluding that the incorporation of the shortcut leads to better results.
(1) He, K., Zhang, X., Ren, S., & Sun, J. (2016). Deep residual learning for image recognition. In Proceedings of the IEEE conference on computer vision and pattern recognition (pp. 770-778).
Point 2: In the Downscaling block module, why is there an activation function after Max Pooling? Please explain why.
Response 2: There was an error in Figure 2, in which we have mistakenly added an activation after the max-pooling, within the downscaling block module. Many thanks for noticing the error, we have edited the figure accordingly, i.e., now it does not have the activation after the max-pooling.
Point 3: In theory, a reasonable loss function should be presented and an explanation of why these functions are used is needed.
Response 3: Given that a proper model loss is crucial to effectively deal with the characteristics of the spatial data which were used, we present, discuss, and test different functions throughout the article. We specifically experimented with a Huber Loss (1), with the premise of attenuating some known issues associated with standard losses such as the RMSE and MAE. In our specific case, our intuition relied on the fact that large error differences would end up having a smaller impact on the final loss value when using the Huber Loss. We compared the Huber Loss against losses based on MAE and RMSE to test this idea, and verified that it produced the best results. Besides this, we also tested the incorporation of different components into the standard Huber Loss, namely with the objective of (i) promoting heterogeneity in the disaggregated results, through the penalization of results with low standard deviation, and (ii) promoting equivariance to spatial transformations, through the maximization of the agreement between original and transformed image representations. These two strategies were tested and validated through different sets of experiments - Tables 7 to 11.
(1) Peter J. Huber "Robust Estimation of a Location Parameter," The Annals of Mathematical Statistics, Ann. Math. Statist. 35(1), 73-101, (March, 1964)
Point 4: The figure of structure of training deep convolutional neural networks are needed to show for some interested readers.
Response 4: The disaggregation method which was used involves two phases. In an initial phase, the aggregated data is translated to the target resolution through the application of a disaggregation heuristic. In a second phase, we iteratively refine these initial estimates through the application of a regression model, which can have the form of a Convolutional Neural Network. Figure 1 specifically details these different steps, by comparing the overall disaggregation framework to the teacher/student framework presented by Xie et al. On the other hand, Figure 2 presents the CNN architecture that we used in our experiments, i.e., the U-Net architecture. It details the different layers that are involved, as well as how the different hyper-parameters of the CNN are used (e.g., the number of feature maps of each layer). Both figures are explained and detailed in the article, together with the clarification of the values used for all the training hyper-parameters. Given this, and in order to complement the mentioned figures with possible additions that can improve the explanation of how the training of the CNN is done, is it possible to clarify or give examples of a “figure of structure of training deep convolutional neural networks”? If the reviewer can further detail this suggestion, we would be glad to incorporate another figure in a second revision (or change the two aforementioned figures accordingly, in order to clarify).
Best regards,
João Monteiro
Reviewer 3 Report
This paper presented a CNN based method for geospatial data disaggregation, in which it explores self-training along with the use of deep neural networks for combining different ancillary data types. The authors introduced the background and context of their research, illustrated the proposed method in details with tables and figures which are easy to read and understand. The experiments were design in a good shape and the results supported the final conclusion. I enjoyed reading the manuscript and would like to recommend the publish.
Author Response
Many thanks for the comments.
Best regards,
João Monteiro
Reviewer 4 Report
It is a technical paper of interest for national readers and appropriate for IJGI. I would see some revisions before acceptance, supposing that the present version is a first revision of an already submitted paper (red sentences...).
- Language usage should be improved further, avoiding longer sentences in the introduction and the final chapters.
- Bibliography is extended enough, but can be improved further in order to clarify the international role of the journal and the present article.
- Originality and novelty of the study should be clarified further in a specific section, hopefully the conclusions.
- I would see a brief comment on further possible research lines.
- Generalization of the approach to broader contexts is welcome. At the moment it is very rough through the manuscript.
Author Response
Many thanks for the comments. We have addressed the following points:
Point 1: Language usage should be improved further, avoiding longer sentences in the introduction and the final chapters.
Response 1: We revised the contents of the paper, attempting to avoid longer sentences, and correcting some typos that we detected during the revision.
Point 2: Bibliography is extended enough, but can be improved further in order to clarify the international role of the journal and the present article.
Response 2: We expanded the bibliography with some additional recent references on spatial data downscaling (including articles published at MDPI journals such as IJGI or “Remote Sensing”), and which share some aspects with the topic of our manuscript. These references complement the claims made in the survey of previous research, or support the additional discussion that was made in Section 6.
Point 3: Originality and novelty of the study should be clarified further in a specific section, hopefully the conclusions.
Response 3: In order to clarify the originality and novelty of the paper, we have added a brief discussion in Section 6 which emphasizes the similarities and differences between our study and those from the literature.
Point 4: I would see a brief comment on further possible research lines.
Response 4: In Section 6, we already present different approaches that can be taken into account for future work, namely (i) the use of other regression models and/or training approaches, (ii) the possibility to weight the different patches when predicting the final results, in order to assign higher importance to those instances likely to be associated to lesser errors, or (iii) the application of an adversarial learning framework that can differentiate between realistic and less plausible patches. To complement these, we also added at the end of Section 6 a brief paragraph on how to use additional loss components for model training, for instance based on computing metrics associated with the patches, or promoting differences between patches computed in self-training iterations far away from each other.
Point 5: Generalization of the approach to broader contexts is welcome. At the moment it is very rough through the manuscript.
Response 5: We have also added a small discussion to Section 6, on the direction that was suggested.
Round 2
Reviewer 1 Report
The author did not answer my questions. It is difficult to see the novelty of this paper because authors fail to give significant differences and new ideas of their work as compared with others.
Author Response
We attempted to provide, in Round 1, answers to the comments and questions that were raised by all the reviewers (including Reviewer 1). The manuscript was also revised accordingly, with the changes over the original version highlighted in red. If the reviewer can further detail his suggestion, we will be glad to provide additional details. The changes that were now introduced in response to the comments from Reviewer 4 also address issues such as discussing the “differences and new ideas of this work as compared with others”, so perhaps some of the concerns from Reviewer 1 have now also been addressed.